# Metabolic engineering of *Escherichia coli* for de novo biosynthesis of vitamin B$_{12}$

Huan Fang [1,2], Dong Li [1], Jie Kang [1], Pingtao Jiang [1], Jibin Sun[1,2] & Dawei Zhang [1,2,3]

The only known source of vitamin B$_{12}$ (adenosylcobalamin) is from bacteria and archaea. Here, using genetic and metabolic engineering, we generate an *Escherichia coli* strain that produces vitamin B$_{12}$ via an engineered de novo aerobic biosynthetic pathway. In vitro and/or in vivo analysis of genes involved in adenosylcobinamide phosphate biosynthesis from *Rhodobacter capsulatus* suggest that the biosynthetic steps from co(II)byrinic acid a,c-diamide to adocobalamin are the same in both the aerobic and anaerobic pathways. Finally, we increase the vitamin B$_{12}$ yield of a recombinant *E. coli* strain by more than ~250-fold to 307.00 µg g$^{-1}$ DCW via metabolic engineering and optimization of fermentation conditions. Beyond our demonstration of *E. coli* as a microbial biosynthetic platform for vitamin B$_{12}$ production, our study offers an encouraging example of how the several dozen proteins of a complex biosynthetic pathway can be transferred between organisms to facilitate industrial production.

[1] Tianjin Institute of Industrial Biotechnology, Chinese Academy of Sciences, Tianjin 300308, China. [2] Key Laboratory of Systems Microbial Biotechnology, Chinese Academy of Sciences, Tianjin 300308, China. [3] National Engineering Laboratory for Industrial Enzymes, Tianjin 300308, China. Correspondence and requests for materials should be addressed to D.Z. (email: zhang_dw@tib.cas.cn)

Strong demand from the food, feed additive, and pharmaceutical industries has recently sparked strong interest in the use of bacteria to produce vitamin $B_{12}$ (adenosylcobalamin)[1]. To date, the industrial production of vitamin $B_{12}$ is mainly from fermentation of *Pseudomonas denitrificans*[2] and *Propionibacterium freudenreichii*[3], but these strains grow slow and are difficult to engineer. Although there are several reports about the genetic engineering of these two strains[4–7], there have not been any reports of the use of genome editing tools with them. Thus, the engineering of the de novo vitamin $B_{12}$ biosynthetic pathway in a fast-growing, genetically tractable species like *E. coli* is attractive as an alternative production platform. However, given that the de novo biosynthetic pathway contains around 30 genes that would need to be expressed heterologously, the successful production of vitamin $B_{12}$ represents a significant engineering challenge.

In nature, vitamin $B_{12}$ is synthesized de novo in some bacteria and archaea using one of two alternative pathways: the aerobic pathway (represented by *P. denitrificans*) and the anaerobic pathway (represented by *Salmonella typhimurium*). The major differences between the two pathways relate to the time of cobalt insertion and the requirement for oxygen[8]. In the aerobic pathway, uroporphyrinogen III (Uro III) is converted to hydrogenobyrinic acid (HBA) via a series of C-methylations and decarboxylations as well as a ring contraction (Fig. 1). Subsequently, HBA is converted to hydrogenobyrinic acid a,c-diamide (HBAD) through amidations of carboxyl groups at positions a and c by the CobB enzyme in *P. denitrificans*[9]. HBAD is converted to co(II)byrinic acid a,c-diamide (CBAD) after cobalt insertion (Catalyzed by the CobNST complex), and CBAD then undergoes co(II)rrin reduction, adenylation, and amidations of carboxyl groups at positions b, d, e, and g to yield adenosylcobyric acid (AdoCby)[10,11]. It has been assumed that adenosylcobinamide (AdoCbi) is synthesized via the attachment of (R)-1-Amino-2-propanol (AP) to AdoCby to yield AdoCbi in a single step reaction that is catalyzed by a two-component system (designated as α and β in *P. denitrificans* (Fig. 1); the β component is known to comprise the CobC and CobD enzymes in this organism, but the α protein component remains unidentified)[9]. AdoCbi then undergoes phosphorylation to form adenosylcobinamide-phosphate (AdoCbi-P) before the final addition of a GMP moiety by the bifunctional enzyme CobP to yield AdoCbi-GDP. In the anaerobic pathway of *S. typhimurium*: AdoCbi-P is biosynthesized through the attachment of (R)-1-Amino-2-propanol O-2-Phosphate (APP) to AdoCby as catalyzed by an AdoCbi-P synthase CbiB (Fig. 1)[12]. APP is synthesized from L-threonine by PduX and CobD[13,14]. AdoCbi-P is transformed into AdoCbi-GDP through the addition of a GMP moiety by CobU[15]. Two additional known reactions transfer lower axial ligands onto AdoCbi-GDP, thus producing adenosylcobalamin (AdoCbl)[16,17].

It has been reported that a recombinant *E. coli* expressing vitamin $B_{12}$ biosynthetic genes from *P. denitrificans* ATCC 13867 produced $0.65 \pm 0.03 \, \mu g \, g^{-1}$ DCW vitamin $B_{12}$[18]; however, this report did not include unambiguous chemical evidence of AdoCbl production, as the microbiological assay used is non-specific for AdoCbl detection and is known to sometimes yield false positives[19]. Further, given that protein α has not been identified in the de novo aerobic vitamin $B_{12}$ biosynthetic pathway, AdoCbi-P (an intermediate of the pathway) was not produced in that study. Thus, to date there has been no demonstration of vitamin $B_{12}$ production in *E. coli*.

Here, seeking to engineer a de novo aerobic vitamin $B_{12}$ biosynthetic pathway in *E. coli*, we needed to determine how AdoCbi-P is synthesized. It also bears mention that there have been no reports CBAD production in a heterologous host, although in vitro assays have been reported previously. These two issues are the key problems that we tackle in this study. By comparing and contrasting in vitro and in vivo CBAD biosynthesis reactions, we determined that cobalt uptake transport proteins are necessary participants for cobalt chelation. We also confirmed that the *bluE*, *cobC*, and *cobD* genes from *R. capsulatus* are functional homologues of, respectively, *pduX*, *cobD*, and *cbiB* from *S. typhimurium*.

We successfully engineer *E. coli* to produce vitamin $B_{12}$ by heterologously expressing a total of 28 genes from *R. capsulatus*, *Brucella melitensis*, *Sinorhizobium meliloti* 320[20], *S. typhimurium*, and *Rhodopseudomonas palustris* that are divided into six engineered modules. The enzymes of Module 1 together produce HBA from Uro III, and Module 2 produces CBAD from HBA. Module 3 includes the four cobalt transport proteins CbiM,N,Q,O enabling cobalt uptake from the environment. Module 4 synthesizes AdoCbi-P from CBAD and L-threonine. Module 5 includes the biosynthetic enzymes CobU, CobS, CobT, and CobC from the endogenous *E. coli* vitamin $B_{12}$ salvage pathway, and is used to convert AdoCbi-P to AdoCbl. Module 6 includes HemO, HemB, HemC, and HemD and is designed as a metabolic engineering booster to generate the aforementioned precursor Uro III to increase HBA biosynthesis. Each module is expressed on individual plasmids or is incorporated into the *E. coli* genome. After metabolic engineering of strains to ameliorate the cobalt chelation and Module 4 bottlenecks, as well as optimization of fermentation conditions, we finally improve the vitamin $B_{12}$ production to $307.00 \, \mu g \, g^{-1}$ DCW.

## Results

**Biosynthesis of CBAD in *E. coli***. We started the construction of the de novo vitamin $B_{12}$ biosynthetic pathway using *E. coli* strain MG1655 (DE3) into which we transformed the pET28-HBA plasmid (Supplementary Data 1) to enable it to synthesize HBA, resulting in strain FH001. FH001 was cultured in 2 × YT medium, and produced $0.73 \, mg \, g^{-1}$ DCW HBA (HPLC analysis). We then expressed the pCDFDuet-1 plasmid harboring *cobB* from *R. capsulatus* in FH001 and thereby generated the HBAD-producing strain FH131 (Supplementary Fig. 1); liquid chromatography/ mass spectrometry (LC-MS) analysis showed that FH131 could produce $0.17 \, mg \, g^{-1}$ DCW HBAD (Fig. 2b).

We conducted a series of in vitro assays to support our identification of functional CobN, CobS, and CobT enzymes for the production of CBAD. To increase the likelihood of obtaining functional versions of these three enzymes, three variants from *S. meliloti*[20], *B. melitensis*, and *R. capsulatus* were tested (total of nine enzymes tested). Note that the substrate HBAD is not commercially available, so we purified it from an HBAD-producing strain via an enzyme-trap method. Previous work has established that purified CobN protein can bind to its substrate HBAD[21], so we expressed the *cobN* genes from *B. melitensis*, *S. meliloti*, and *R. capsulatus* with N terminal fusion hexa-histidine tags using the pCDF-cobB plasmid and transformed these into the FH001 strain, respectively generating strains FH159, FH160, and FH161. Cultures of these strains were used to purify CobN and HBAD complexes via the enzyme-trap method detailed in the Methods section, and after boiling, LC-MS analysis showed that HBAD could be detected in all samples (Supplementary Fig. 2), thereby demonstrating that CobN can bind its substrate HBAD without the aid of CobS or CobT.

We next attempted to purify CobS and CobT from *S. meliloti*, *B. melitensis*, and *R. capsulatus*. CobS and CobT from *S. meliloti*, *B. melitensis*, or *R. capsulatus* (each with a C-terminal hexa-histidine tag) were purified via affinity chromatography (SDS-PAGE analysis shown in Supplementary Fig. 3). We conducted in vitro assays with HBAD, CobN, CobS, and CobT and used LC-

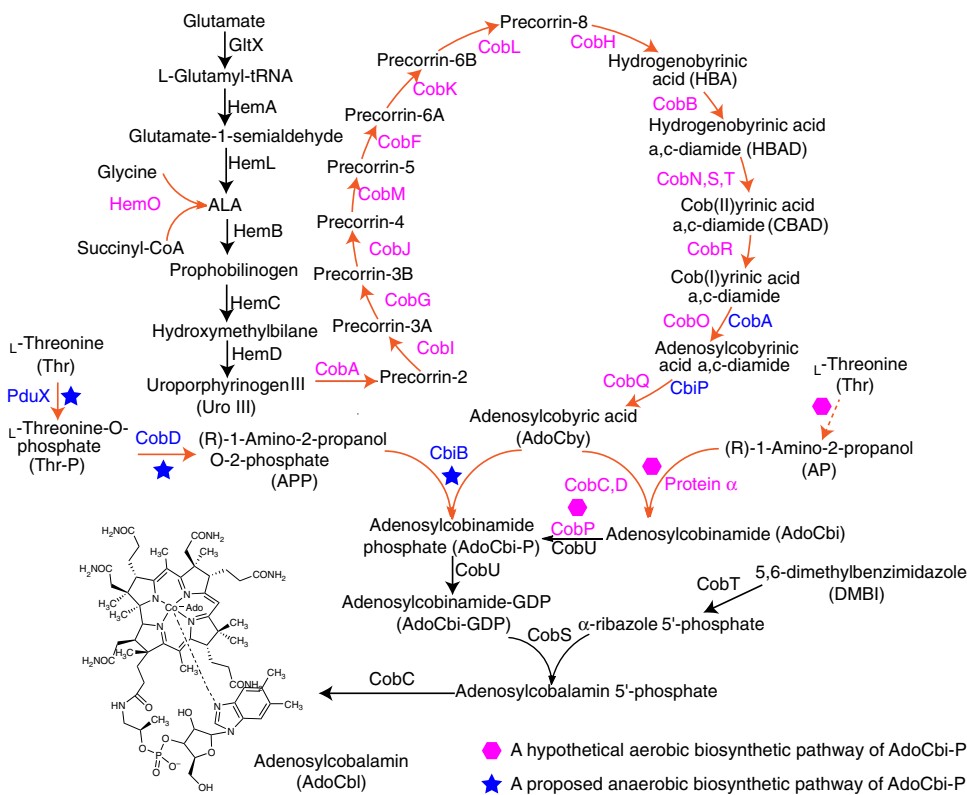

**Fig. 1** Biosynthetic pathway of adenosylcobalamin. Endogenous enzymes from *E. coli* are shown in black. Enzymes from aerobic bacteria such as *B. melitensis*, *R. capsulatus*, *S. meliloti*, and *R. palustris* are shown in magenta. Enzymes from *S. typhimurium* are shown in blue. Ado represents the abbreviation of adenosyl

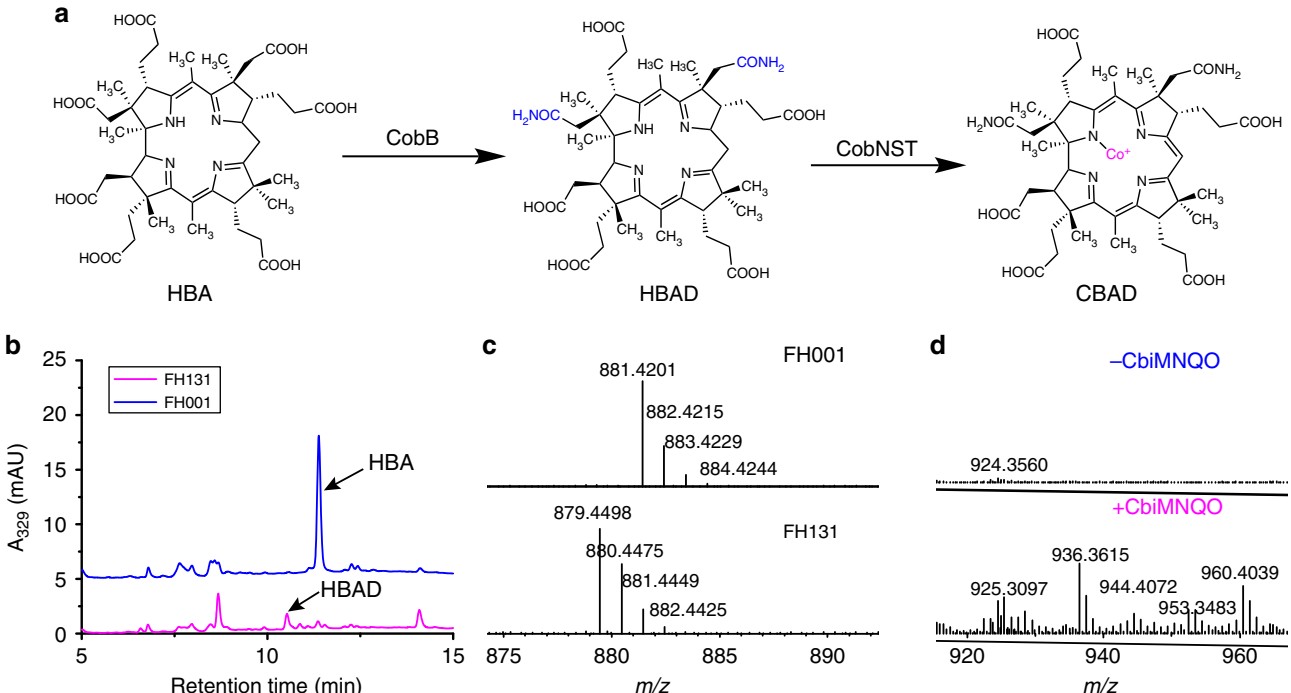

**Fig. 2** The CBAD biosynthetic pathway and LC–MS detection of biosynthetic intermediates. **a** HBA is converted to CBAD by CobB and the CobN,S,T enzymes. **b** The strain FH131 expressing *cobB* converted HBA to HBAD, as assessed by HPLC. **c** Comparison of mass spectra for HBAD from the FH001 and FH131 stains. **d** Comparison of mass spectra for CBAD from recombinant *E. coli* stains with or without the expression of the *cbiM,N,Q,O* proteins

MS to confirm their activity: CBAD was formed when the substrate HBAD, the three enzymes, and cobalt chloride were all present, but was not detected in the cotrol group lacking cobalt chloride (Supplementary Figs. 4, 5, 6 and Supplementary Note 1).

Having confirmed the ability of CobN, CobS, and CobT to produce CBAD in vitro, we next expressed these proteins in vivo to synthesize CBAD. Initially we used the *cobB* gene from *R. capsulatus* and *cobN* (with a fused N-terminal hexa-his tag to increase translation efficiency; SDS-PAGE analysis shown in Supplementary Fig. 7), *cobS*, and *cobT* genes from *S. melloti, B. melitensis* or *R. capsulatus* (expressed as pseudo-operons) on a single plasmid (Module 2). LC-MS analysis of these strains did not detect CBAD, and these discrepencies in the results from the in vivo vs. in vitro assays prompted us to consider how cobalt metabolism may have affected biosynthesis in our *E. coli* cells. It has been previously reported that the heterologous expression in *E. coli* of the CbiM,N,Q,O proteins from *S. typhimurium* or from *R. capsulatus* enables cobalt uptake[22]. To this end, a plasmid carrying genes for these four proteins (Module 3) was co-transformed into the FH001 strain alongside plasmids harboring Module 2, generating strains FH163, FH164, and FH165 (a separate strain for each of the Module 2 variants, see above).

In agreement with our hypothesis, each of these strains could produce CBAD, demonstrating the necessity of cobalt uptake transporters for cobalt chelation (Fig. 2). FH164 had the highest CBAD yield among these three strains. Module 3 was then integrated into the *E. coli* genome and the *endA* gene (encoding DNA-specific endonuclease I) was deleted, resulting in strain FH215. When the plasmids pET28-HBA (Module 1) and pCDF-cobB-BmcobN-his-BmcobS-BmcobT (Module 2) were transformed into FH215, CBAD could be produced in the new strain FH216, which was subsequently used as the CBAD-producing strain for our continued engineering of vitamin B$_{12}$ production in *E. coli*.

**Biosynthesis of APP in *E. coli*.** A pathway for APP—an intermediate in cobalamin biosynthesis—has been proposed in *S. typhimurium*: L-threonine undergoes phosphorylation by PduX and decarboxylation by *St*CobD (CobD from *S. typhimurium*) (Fig. 3a). APP is then condensed with AdoCby into AdoCbi-P by CbiB. Although the functions of PduX and *St*CobD have been demonstrated in vitro, the supposed APP biosynthetic pathway has not been verified in vivo. In *P. denitrificans*, it is recognized that AP is somehow attached to AdoCby catalyzed by a complex comprising CobC, CobD, and the as-yet-unidentified protein α. However, the Km value for AP was very high (20 mM[23]), suggesting that AP might not be the true substrate for this AP-synthesizing complex. Interestingly, we found that homologues of the anaerobic pathway genes *pduX*, *cobD*, and *cbiB* from *S. typhimurium* are present in genome of the aerobic-pathway-utilizing *R. capsulatus* as, respectively, the *blue*, *cobC*, and *cobD* genes.

It thus appears likely that the aerobic and anaerobic pathways employ the same biosynthetic reactions to produce APP and AdoCbi-P. To test this hypothesis, *blue* and *cobC* from *R. capsulatus*, which encode potential L-threonine kinase and threonine-O-3-phosphate decarboxylase, respectively, were assayed in vitro and in vivo. BluE and PduX were initially expressed at a very low level from the pACYCduet-1 plasmid. To minimize locally stabilized mRNA secondary structures and thereby enable efficient translation initiation of the *bluE* and *pduX* transcripts, two bicistronic vectors were designed. In the first plasmid (pACYC-MBP-bluE), a sequence derived from the first ten amino acids of the maltose binding protein (MBP) was placed located upstream of the second codon-optimized *bluE*

sequence (Supplementary Fig. 8a, b). In the second plasmid (pACYC-his-pduX), a sequence derived from a his tag is located upstream of the second *pduX* seqcuence (Supplementary Fig. 8c, d). SDS-PAGE analysis showed that the expression level of *bluE* and *pduX* were significantly improved in strains carrying this system (Fig. 3b). As BluE degenerated almost completely during pufication, crude BluE activity was confirmed via an in vitro assay (Supplementary Fig. 9). The activity of *Rc*CobC (CobC from *R. capsulatus*) was also confirmed via in vitro assays (Supplementary Fig. 10). The Km and Vmax values of *Rc*CobC were measured to be 1.89 mM and 25.6 μM/min, respectively.

Given that *E. coli* has an endogenous pathway to produce AP (Fig. 3a), we deleted the *gldA* gene from *E. coli* MG1655 (DE3); the resulting strain was designated as FH291. We then designed and introduced an AP pathway into FH291 to confirm the functions of BluE/PduX and *Rc*CobC/*St*CobD in vivo (Fig. 3c–e): the new BluE/PduX strains FH296/FH298 both produced L-threonine O-3-phosphate (Thr-P) (HPLC analysis), demonstrating that both PduX from *S. typhimurium* and BluE from *R. capsulatus* are L-threonine kinases; The new *Rc*CobC/*St*CobD strains FH297/FH299 both produced AP, demonstrating that CobD from *S. typhimurium* and CobC from *R. capsulatus* are L-threonine-O-3-phosphate decarboxylases. These results established that functional BluE/PduX and *Rc*CobC/*St*CobD can be used to synthesize APP in recombinant *E. coli* strains. We also deleted the genes encoding PhoA (an alkaline phosphatase) and AphA (an acid phosphatase) successively in FH297 and FH299. However, as each of these strains still produced AP, the phosphatase that converts APP to AP in *E. coli* is not yet clear.

**De novo biosynthesis of vitamin B$_{12}$ in *E. coli*.** AdoCby is formed from CBAD in three steps. In the first step, CBAD is reduced to cob(I)yrinic acid a,c-diamide by an NADH-dependent flavoenzyme that (in aerobic *Pseudomonas denitrificans*) exhibits cob(II)yrinic acid a,c-diamide reductase activity. However, the gene encoding this enzyme has not been identified. The *cobR* gene from aerobic *B. melitensis* has been demonstrated to encode a biosynthetic enzyme with co(II)rrin reductase activity[24]. In the next biosynthesis step of the known pathway, cob(I)yrinic acid a, c-diamide is transformed to its adenosyl form by an adenosyl-transferase. In some bacteria, identical adenosyltransferases catalyze adenosylation reactions on cobinamide in a salvage pathway. Therefore, the adenosyltransferase BtuR of *E. coli* in the salvage pathway might function on the substrate cob(I)yrinic acid a,c-diamide.

To explore this idea, the strain FH216 was used as the host cell for functional testing of a Cbi module (Module 4). Seeking to synthesize the precursor AdoCbi-P, the design of this module included both proteins of known function (CobR from *B. melitensis* and AdoCby synthase and AdoCbi-P synthase from *S. typhimurium*, as well as a L-threonine kinase and a threonine-O-3-phosphate decarboxylase from each of *S. typhimurium* and *R. capsulatus*) and candidate enzymes (BtuR from *E. coli* and an AdoCbi-P synthase from *R. capsulatus*). Note that *E. coli* can assemble a nucleotide loop (modified lower axial ligand) using endogenous CobU, CobT, CobS, and CobC, so AdoCbi-P can be transformed into AdoCbl using the endogenous *E. coli* salvage pathway[12,25].

Finally, two recombinant strains FH218 (with *pduX*, *cobD*, and *cbiB* from *S. typhimurium*) and FH219 (with *bluE*, *cobC*, and *cobD* from *R. capsulatus*) harboring Modules 1–5 were obtained (Fig. 4a). We supplemented the CM medium with L-threonine to circumvent the need to engineer the supply of this amino acid in host cells. Our dectection of vitamin B$_{12}$ by LC-MS and comparison of its retention time and spectrum with the chemical

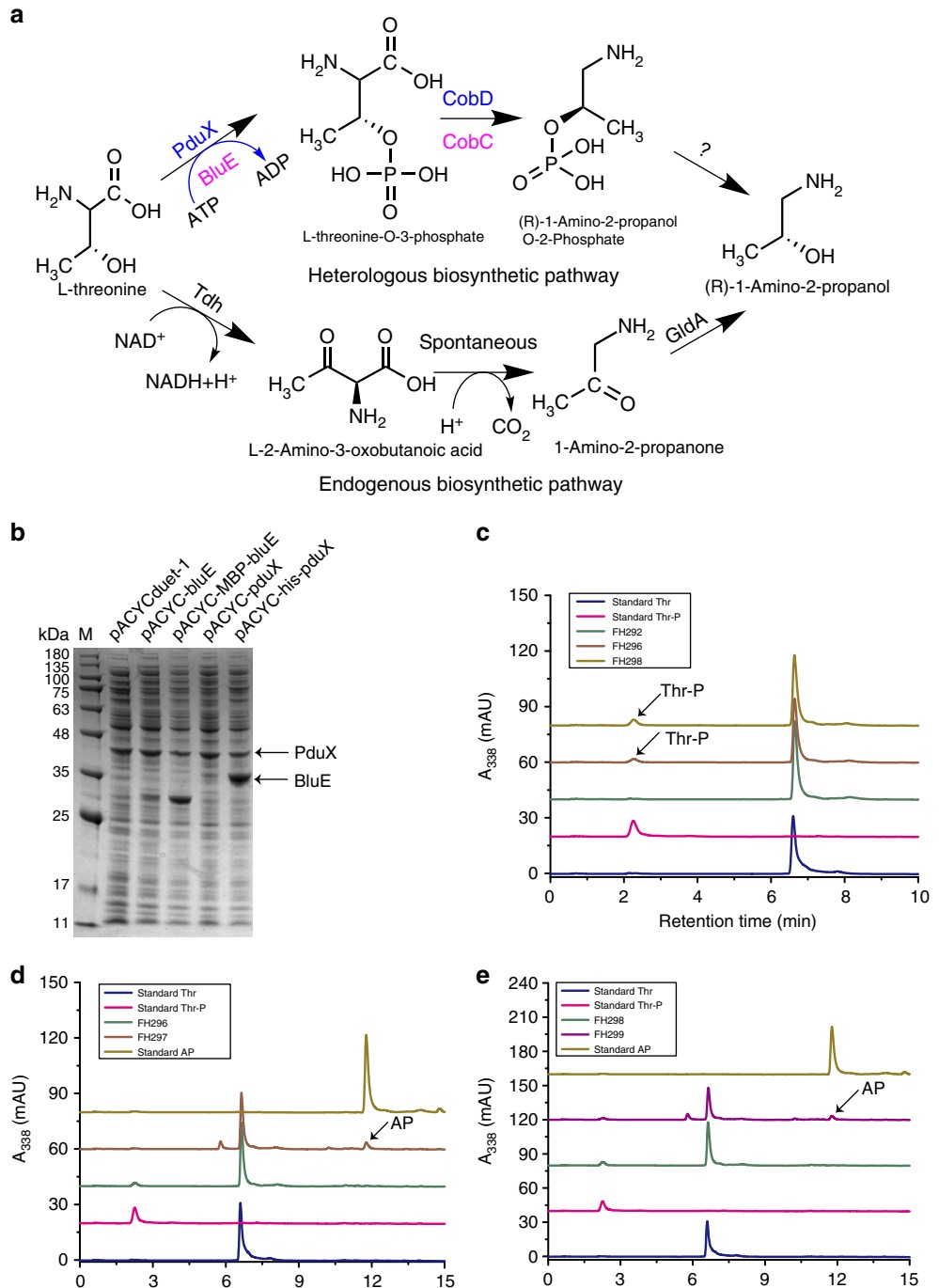

**Fig. 3** Metabolic engineering of the heterologous APP biosynthetic pathway in *E. coli*. **a** Endogenous and heterologous APP biosynthetic pathway in *E. coli*. **b** Optimization of the expression of the *blue* and *pduX* genes via a bicistronic design that included an MBP tag upstreat of the *bluE* gene and a his tag upstream of the *pduX* gene. **c** Confirmation of the enzymatic functions of PduX and BluE in *E. coli*. **d** Confirmation of function of CobD from *S. typhimurium* in *E. coli*. **e** Confirmation of the function of CobC from *R. capsulatus* in *E. coli*. Note that Calf intestinal alkaline phosphatase can transform APP to AP. Owing to the lack of a commercially available APP standard, we indirectly monitored the APP level by detecting conversion to AP using the endogenous phosphatase of *E. coli*

reference standard demonstrated the successful assembly of our completed engineered de novo vitamin B$_{12}$ biosynthesis pathway in recombinant strains FH218 and FH219. Beyond this bioengineering result, our successful demonstration of vitamin B$_{12}$ in this system confirms the scientific fact that APP participates in the biosynthesis of AdoCbi-P in both the aerobic and anaerobic pathways (Fig. 4). However, FH218 and FH219 produced only 2.18 µg g$^{-1}$ DCW and 1.22 µg g$^{-1}$ DCW vitamin B$_{12}$, respectively (Supplementary Fig. 11), and it should be noted that HBA was not detected in FH164, a finding which suggested the idea that HBA availability may be a limit to CBAD synthesis.

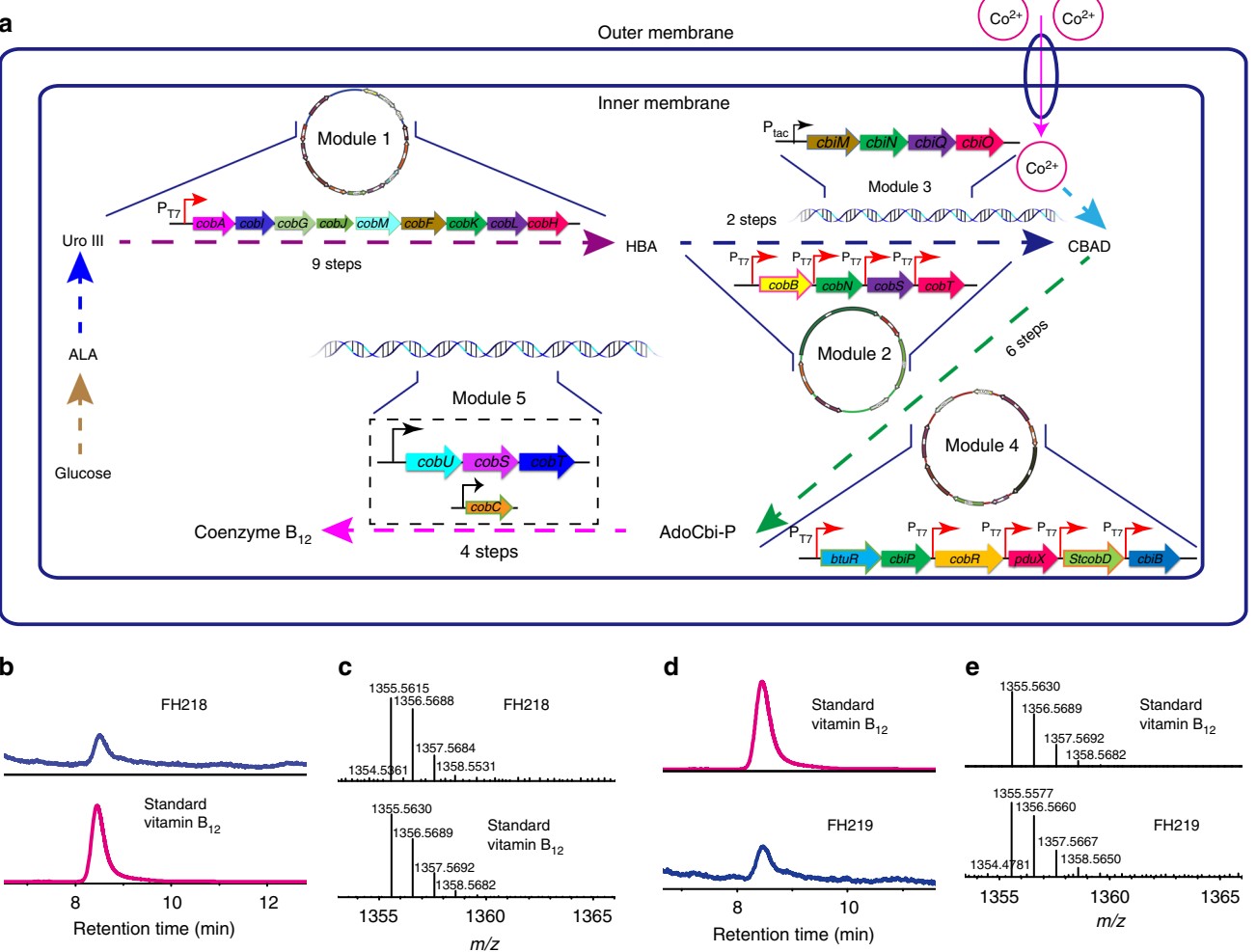

**Fig. 4** Metabolic engineering of vitamin $B_{12}$ production in *E. coli*. **a** Metabolic engineering in *E. coli* of a de novo vitamin $B_{12}$ biosynthetic pathway. Five modules are used to synthesize vitamin $B_{12}$ from glucose in *E. coli*. **b** HPLC analysis and **c** mass spectrometry analysis of vitamin $B_{12}$ produced by the FH218 strain. **d** HPLC analysis and **e** mass spectrometry analysis of vitamin $B_{12}$ produced by FH219. Adenosylcobalamin was converted to vitamin $B_{12}$ for detection

**Increasing the HBA pool increased vitamin $B_{12}$ production.**
Vitamin $B_{12}$ and other tetrapyrrole compounds including heme, siroheme, and coenzyme $F_{430}$ each originate from δ-aminolevulinate (ALA)[26]. ALA is synthesized via either a $C_4$ or $C_5$ pathway[27,28]. In the $C_4$ pathway, glycine and succinyl-CoA are condensed by ALA synthase (encoded by either *hemA* or *hemO*) to form ALA. In the $C_5$ pathway, ALA is synthesized from glutamate via a three-step reaction catalyzed successively by the enzymes Gltx, HemA, and HemL[29]. Then, eight molecules of ALA undergo condensation, polymerization, and cyclization to form Uro III, which is a precursor for the biosynthesis of vitamin $B_{12}$, heme, and coenzyme F430[2].

Given that the *cobAIGJMFKLH* operon is driven by a strong T7 promoter, we heterologously expressed the *hemO*, *hemB*, *hemC*, and *hemD* genes (the pathway components from ALA to Uro III) with the aim of converting additional precursor molecules into HBA. HemO from *R. palustris* and HemB, HemC, and HemD from *S. meliloti* were expressed as a polycistronic product in the FH001 strain, which boosted HBA production from 0.54 mg g$^{-1}$ DCW to 6.43 mg g$^{-1}$ DCW after 20 h of growth, albeit with reduced overall growth (Supplementary Fig. 12). Our experiments showed that HBA production in the FH001 strain gradually decreased from 8 to 38 h. However, HBA production of FH168 peaked at 20 h and then gradually decreased, highlighting that the

accumulation of precursors from ALA to Uro III does boost HBA production in FH168. We speculate that this apparent gradual decrease in HBA levels from 20 to 38 h may related to HBA stability or perhaps the decreased catalytic function of the heterologous enzymes over time.

In addition, the maximum ALA yields for the FH001 and FH168 strains were 0.10 and 0.67 g l$^{-1}$, respectively, confirming that the expression of heterologous ALA synthase boosted ALA production. Expression of a polycistronic product from a *hemOBCD* gene cluster lead to heme accumulation and was toxic[30]. To decrease the formation of heme, we used a small RNA strategy[31] to knockdown the expression of *hemE*, *hemF*, *hemG*, and *hemH* in FH168 (Supplementary Fig. 13). The FH187 strain (*hemG* knockdown) had the highest HBA yield at 9.34 mg g$^{-1}$ DCW, a 45.26% increase over FH168. We then assessed six strains that had the expression of two of their heme biosynthesis genes simultaneously knocked down via small RNA (Supplementary Fig. 14). Of particular note, the HBA yield of the strain FH192 (*hemF* and *hemG* knockdown) was 14.09 mg g$^{-1}$ DCW, a 50.86% increase over FH187. Collectively, these experiments established that the heterologous expression of a Uro III module (Module 6) combined with the knockdown of heme biosynthetic gene expression can boost HBA production in recombinant *E. coli*.

Next, seeking to reduce the number of plasmids/strain, Module 6 was integrated into the $P_{BAD}$ locus of the *E. coli* genome, resulting in FH236. The plasmid pET28-HBA-antihemFG (Module 1 + dual heme gene knockdown) was transformed into FH236, generating strain FH322, which accumulated 2.00 mg g$^{-1}$ DCW HBA. Adding CobB produced strain FH273, which accumulated 0.43 mg g$^{-1}$ DCW HBAD. The introduction of plasmids carrying Module 2 and 4 were both transformed into FH322, generating FH309, which produced 5.72 μg g$^{-1}$ DCW vitamin B$_{12}$ (compare with 2.18 μg g$^{-1}$ DCW for FH218 and 1.22 μg g$^{-1}$ DCW for FH219).

**Genetic modifications of Modules 2 and 4**. The gap between the production of HBAD of FH273 and vitamin B$_{12}$ of FH309 implied that metabolic bottlenecks might exist among cobalt chelatases and Cbi module. In addition, HBAD accumulated in FH309 (Supplementary Fig. 15), suggesting that the cobalt chelation step is indeed a bottleneck. Therefore, recombinant strains expressing cobalt chelatases from *B. melitensis*, *S. meliloti*, or *R. capsulatus* were evaluated for vitamin B$_{12}$ production (Fig. 5a). The strain with the cobalt chelatase from *R. capsulatus* (FH329) had highest vitamin B$_{12}$ production among these strains, at 21.96 μg g$^{-1}$ DCW. In addition, six recombinant *E. coli* strains variously expressing combinations of the CobN, CobS, and CobT enzymes from different bacterial species were also evaluated for vitamin B$_{12}$ production. These strains produced vitamin B$_{12}$ in amounts ranging from 2.19 μg g$^{-1}$ DCW to 11.22 μg g$^{-1}$ DCW (Fig. 5b).

A review article has speculated that the enzyme CobW may somehow function in the delivery and presentation of cobalt to CobN[32], so we investigated how the expression of CobW might improve vitamin B$_{12}$ production. Interestingly, vitamin B$_{12}$ production increased in five of the nine recombinant strains in which we expressed various *cobW* homologs (Fig. 5a, b). The strain FH351 expressing *cobW* from *B. melitensis* achieved the highest vitamin B$_{12}$ production, at 68.61 μg g$^{-1}$ DCW, a 14.9-fold increase over the original strain lacking *cobW*.

After optimizing the cobalt chelatases, we tested a multiple variants of the downstream Cbi module with the aim of further improving vitamin B$_{12}$ production. Most of the metabolic intermediates between CBAD and AdoCbl are instable and unavailable commercially, which limited our ability to conduct in vitro enzyme kinetics assays or to probe each step in vivo to identify specific bottlenecks. It is well-established that the expression of enzyme variants can help to increase metabolic flux toward traget products[33]. Thus, Cbi module variants comprising enzymes from various bacteria were examined (Fig. 5c), and among the strains expressing CobW, strain FH364 (expressing *cobR* from *B. melitensis*, StcobA, cbiP, pduX, StcobD, and *cbiB* from *S. typhimurium*) had the highest vitamin B$_{12}$ production at 171.81 μg g$^{-1}$ DCW. It is noteworthy that three strains harboring the *bluE-RccobC-cbiB* expression cassette (FH362, FH363, and FH367) could synthesize vitamin B$_{12}$, demonstrating that BluE and CobC from *R. capsulatus* perform the same catalytic functions as do PduX and CobD from *S. typhimurium* in the de novo vitamin B$_{12}$ biosynthesis pathway. In addition, the three strains harboring the *pduX-StcobD-RccobD* expression cassette (FH370, FH371, and FH372) could also synthesize vitamin B$_{12}$, demonstrating that CobD from *R. capsulatus* performs the same catalytic function as does CbiB from *S. typhimurium* in the de novo vitamin B$_{12}$ biosynthesis pathway. These results further highlight that aerobic bacteria and anaerobic bacteria both use the same pathway to synthesize AdoCbi-P from AdoCby.

We also investigated whether regulating copy numbers of plasmids could further improve vitamin B$_{12}$ production. The plasmids harboring Modules 1, 2, and 4 contained the respective replicons pMB1 (medium copy numbers), CloDF13 (high copy numbers), and p15A (medium copy numbers). As the amount of HBA is sufficient for these recombinant *E. coli* strains, the replicon of this plasmid was not changed, while replicons of plasmids harboring Module 2 and 4 were changed, resulting in five strains harboring compatible plasmids with different copy numbers (Supplementary Table 1). We observed a decrease in vitamin B$_{12}$ production when the replicon of the plasmid harboring Module 3 was changed to the low copy number pSC101 (139.56 μg g$^{-1}$ DCW vs. 171.81 μg g$^{-1}$ DCW for FH364). In addition, when replicons of the plasmid harboring Module 2 were changed to the medium copy p15A or low copy pSC101, vitamin B$_{12}$ was not detected in the resulting four strains. These findings indicate that Module 2 can be a bottleneck if medium or low copy plasmids are used. Finally, these results also indicate the top-performing strains should have plasmids containing Modules 1, 2, and 4 with medium, high, and medium copy numbers, respectively.

Plasmid stability assays of the strain FH364 indicated these plasmids are very stable when FH364 is cultured without isopropyl-β-D-thiogalactoside (IPTG) induction during fermentation. The plasmid containing Module 1 was very stable, even when cultured for 20 h after IPTG induction. However, the other two plasmids showed instablity after IPTG induction, indicating that a high metabolic burden, but not the plasmid itself, led to plasmid instability. Specifically, the plasmids containing Modules 2 and 4 in FH364 were gradually lost during fermentation: 47.34% and 36.06%, of these respective plasmids were lost when cultured for 20 h after IPTG induction (Supplementary Table 2). Plasmid stability assays of the five strains harboring plasmids with different copy numbers suggested that plasmids with high copy numbers caused plasmid instability, while plasmids with medium and low copy numbers in FH381 and FH383 were very stable (Supplementary Table 2). We speculate that both increased plasmid stability and improved maintenance of plasmid copy number could be achieved by expressing essential genes on the plasmids in future efforts.

**Optimization of fermentation conditions**. In order to provide an optimal growth medium for the engineered strains, the concentrations of key precursors such as glycine, succinic acid, and betaine were optimized in the medium (Supplementary Table 3 and Supplementary Fig. 16). Vitamin B$_{12}$ production of FH364 increased to 255.68 μg g$^{-1}$ DCW when cultured in the optimized modified CM medium. We then altered growth temperatures and IPTG induction concentrations, as these are known to influence heterologous protein expression (Supplementary Table 4 and Supplementary Fig. 17). We found that growth temperature has a greater influence on vitamin B$_{12}$ production than IPTG concentrations: low temperature is conducive to the expression of soluble proteins and to correct folding, while high temperature is conducive to bacteria growth. Vitamin B$_{12}$ production increased from 24 to 32 °C, and then declined sharply at 37 °C. Cell growth rates increased slightly from 24 to 30 °C, but decreased from 30 to 37 °C. Induction at 0.1 mM IPTG resulted in the lowest vitamin B$_{12}$ production observed for each temperature, and we found that the optimal IPTG concentrations were different for each culture temperature. Collectively, we found that the optimal conditions for is maximal vitamin B$_{12}$ production is a growth temperature of 32 °C and an IPTG concentration of 1 mM: under these optimized conditions the vitamin B$_{12}$ yield of the top-performing FH364 strain increased to 307.00 μg g$^{-1}$ DCW (equal to 0.67 mg l$^{-1}$).

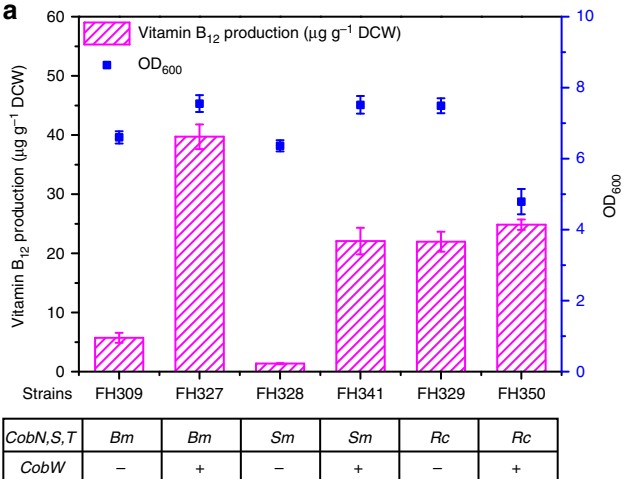

| CobN,S,T | Bm | Bm | Sm | Sm | Rc | Rc |
|---|---|---|---|---|---|---|
| CobW | − | + | − | + | − | + |

Strains: FH309, FH327, FH328, FH341, FH329, FH350

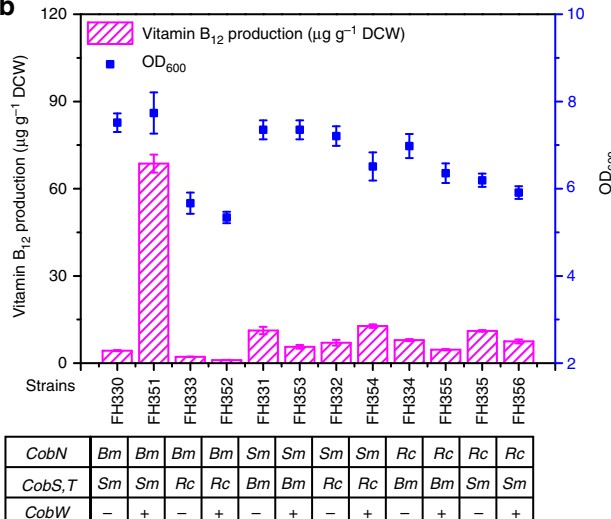

| CobN | Bm | Bm | Bm | Bm | Sm | Sm | Sm | Sm | Rc | Rc | Rc | Rc |
|---|---|---|---|---|---|---|---|---|---|---|---|---|
| CobS,T | Sm | Sm | Rc | Rc | Bm | Bm | Rc | Rc | Bm | Bm | Sm | Sm |
| CobW | − | + | − | + | − | + | − | + | − | + | − | + |

Strains: FH330, FH351, FH333, FH352, FH331, FH353, FH332, FH354, FH334, FH355, FH335, FH356

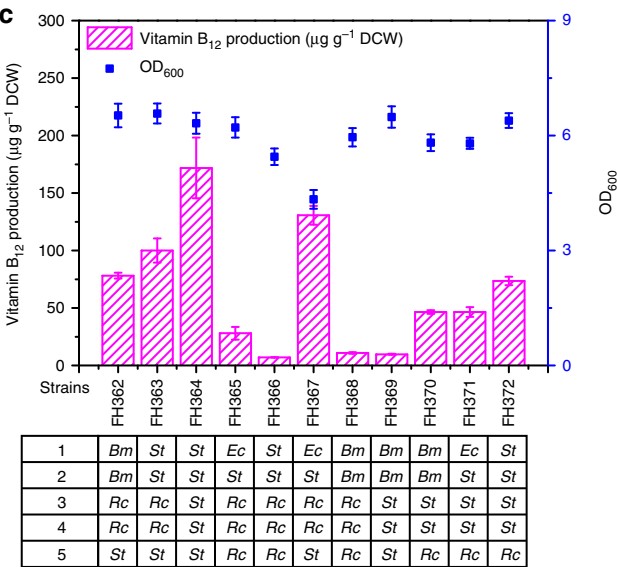

| 1 | Bm | St | St | Ec | St | Ec | Bm | Bm | Bm | Ec | St |
|---|---|---|---|---|---|---|---|---|---|---|---|
| 2 | Bm | St | St | St | St | St | Bm | Bm | Bm | St | St |
| 3 | Rc | Rc | St | Rc | Rc | Rc | Rc | St | St | St | St |
| 4 | Rc | Rc | St | Rc | Rc | Rc | Rc | St | St | St | St |
| 5 | St | St | St | Rc | Rc | St | Rc | St | Rc | Rc | Rc |

Strains: FH362, FH363, FH364, FH365, FH366, FH367, FH368, FH369, FH370, FH371, FH372

**Fig. 5** Comparison of vitamin B$_{12}$ production and growth in *E. coli* strains expressing Module 4 and 5. **a** Vitamin B$_{12}$ production and growth of *E. coli* expressing the *cobN,S,T* genes from a single species (*B. melitensis*, *S. meliloti* or *R. capsulatus*) as well as the *cobW* gene from *B. melitensis*, *S. meliloti*, or *R. capsulatus*. **b** Vitamin B$_{12}$ production and growth of *E. coli* strains expressing various combinations of the *cobN*, *cobS*, *cobT*, and *cobW* genes from *B. melitensis*, *S. meliloti*, or *R. capsulatus*. **c** Vitamin B$_{12}$ production and growth of *E. coli* expressing different variant forms of Module 4. *Bm*, *Sm*, *Rc*, *St*, and *Ec* represent abbreviations of strains *B. melitensis*, *S. meliloti*, *R. capsulatus*, *S. typhimurium*, and *E. coli*, respectively. 1, 2, 3, 4, and 5 in Fig. 5c represent genes encoding cob(I)yrinic acid a,c-diamide adenosyltransferase, adenosylcobyric acid synthase, L-threonine kinase, threonine-O-3-phosphate decarboxylase, and AdoCbi-P synthase, respectively. All strains were cultured in CM medium for vitamin B$_{12}$ production. Error bars indicate standard deviations from triplicate biological replicates

Seeking to replace the traditionally used industrial strains for vitamin B$_{12}$ production, we here chose *E. coli* as the heterologous host to introduce an engineered de novo aerobic vitamin B$_{12}$ biosynthetic pathway. Most of the intermediate molecules of the vitamin B$_{12}$ biosynthetic pathway are not commercially available, which complicates the verification of the functions of candidate enzymes, so we used a bottom-up strategy that integrating genetic and metabolic engineering strategies to develop our engineered vitamin B$_{12}$-producing *E. coli* strain. Based on rational knowledge of the aerobic pathway, we divided the 32 genes known to be involved in the pathway into six separate modules, which facilitated our characterization of each aspect of the design. Ultimately, our efforts led to the first demonstration of complete vitamin B$_{12}$ production in *E. coli*.

It has been reported that the *cobN*, *cobS*, and *cobT* genes from *B. melitensis* and *P. denitrificans* can produce CBAD production in vitro[21,34]. However, it is understood that the mechanism of cobalt chelation in vivo is more complicated than that in vitro[21,34]. We here characterized the role of cobalt transporters in facilitating cobalt chelation. We used in vitro assays to successfully confirm the CBAD-production-related functions of CobN, CobS, and CobT enzymes, but found that the expression of the genes encoding these enzymes in *E. coli* did not result in CBAD production in cells, ostensibly because these cells lacked the essential substrate cobalt. Consistent with this idea, the expression in these cells of the cobalt uptake transport proteins CbiM, CbiN, CbiQ, and CbiO led to CBAD production, thus establishing that having a cobalt uptake transport system is essential for de novo vitamin B$_{12}$ biosynthesis using the aerobic pathway in *E. coli*. This may be a requirement for vitamin B$_{12}$ biosynthesis in all bacteria: consider for example that an ABC-type cobalt transport system (CbtJKL) is essential for growth of *S. meliloti* at trace metal concentrations[35] and that the B$_{12}$ biosynthesis superoperon of *S. typhimurium* includes the *cbiMNQO* gene cassette[19].

Importantly, our study overturned previous assumptions about the aerobic pathway that AP is condensed with AdoCby to form AdoCbi before undergoing a subsequent phosphorylation catalyzed by CobP. Our in vitro and in vivo assays of BluE and *Rc*CobC demonstrated that these enzymes function as, respectively, a L-threonine kinase and a threonine-O-3-phosphate decarboxylase. Expression of CobD from *R. capsulatus* or CbiB from *S. typhimurium* can both facilitate APP's condensation with AdoCby to form AdoCbi-P. It is not AdoCbi that undergoes phosphorylation catalyzed by CobU from *S. typhimurium* or by CobP from the aerobic-pathway-utilizing *R. capsulatus*; rather, APP is condensed to AdoCby to yield AdoCbi-P. These results

## Discussion

The biosynthetic pathway of vitamin B$_{12}$ is very complex and although most of the biosynthetic pathway steps have been elucidated, the process through which AdoCbi-P is synthesized in the aerobic vitamin B$_{12}$ biosynthetic pathway remains unknown.

strongly suggest that aerobic bacteria and anaerobic bacteria may use the same de novo pathway steps from CBAD to AdoCbl.

Our engineered de novo synthetic pathway for vitamin $B_{12}$ was successfully constructed in *E. coli* by assembling six modules comprising many biosynthetic enzymes and four cobalt uptake transport proteins. Although *E. coli* is a kind of facultative anaerobic bacterium, our study demonstrated that it can be modified to synthesize vitamin $B_{12}$ using the aerobic pathway. The original recombinant stains produced around $1–2\,\mu g\,g^{-1}$ DCW of vitamin $B_{12}$. Subsequently, by strengthening the biosynthesis of precursors and making genetic modifications to cobalt chelatases and Cbi module, we were able to increase the vitamin $B_{12}$ production to $171.81\,\mu g\,g^{-1}$ DCW. It was interesting to note that the expression of *cobW* (strain FH351) increased vitamin $B_{12}$ productivity by 14.9-fold, highlighting that *cobW* plays an important role in cobalt chelation. Vitamin $B_{12}$ production in FH364 was finally increased to $307.00\,\mu g\,g^{-1}$ DCW (equal to $0.67\,mg\,l^{-1}$) after optimization of culture media recipe, growth temperature, and IPTG concentrations; this final yield was 250.64-fold higher than the original strain bearing our engineered de novo vitamin $B_{12}$ biosynthetic pathway.

Although vitamin $B_{12}$ productivity of the engineered *E. coli* strain is much less than industrial strains such as *P. denitrificans*[36] (the highest vitamin $B_{12}$ production of $214.3\,mg\,l^{-1}$) and *P. freudenreichii*[37] (the highest vitamin $B_{12}$ production of $206.0\,mg\,l^{-1}$), it is comparable with some native vitamin $B_{12}$ producers, for example: wild-type *Bacillus megaterium* and an engineered strain produced 0.26 and $8.51\,\mu g\,l^{-1}$ vitamin $B_{12}$, respectively[38], and wild-type *P. denitrificans* produced $2.75\,\mu g\,g^{-1}$ DCW vitamin $B_{12}$[18]. Consider that the vitamin $B_{12}$ production of a single *P. denitrificans* strain was increased approximately 100-fold over a 10-year period via a great many rounds of random mutagenesis[4]; however, the vitamin $B_{12}$ production of the *E. coli* strain we engineered in the present study was increased about 250-fold in several months, without random mutagenesis, and can very likely be improved further. Our engineered vitamin $B_{12}$ producing *E. coli* strain overcomes some major limitations that currently face industrial producers of vitamin $B_{12}$ like long growth cycles. For example, the present industrial vitamin $B_{12}$ producing *P. denitrifians* strain needs about 180 h to complete fermentation[36,39,40], whereas the recombinant *E. coli* strain constructed here only needs 24 h. Our successful demonstration of the engineering of the complete pathway for vitamin $B_{12}$ production *E. coli* offers a powerful illustration for how the dozens of genes of a complex heterologous pathway can be successfully introduced into new host organisms. This demonstration of our modular design concept should offer encouragement to other scientists and biotechnologists that are considering similar undertakings.

## Methods

**Chemicals**. Q5 High-Fidelity DNA polymerase and calf intestinal alkaline phosphatase were purchased from New England Biolabs (USA). Taq PCR Master Mix and DNA ladder were from Tiangen (China). Restriction endonucleases, T4 DNA ligase, and T4 Polynucleotide Kinase (PNK) were purchased from Thermo Fisher Scientific (USA). Chromatography-grade methanol and formic acid were from Thermo Fisher Scientific (USA). ALA, (R)-1-Amino-2-propanol, L-threonine O-3-phosphate, and vitamin $B_{12}$ were from Sigma-Aldrich (USA). Ni Sepharose 6 Fast Flow was from GE Healthcare (USA). DEAE-Sephadex A25 was from Pharmacia (USA).

**Medium and growth conditions**. For routine purposes, *E. coli* derived strains were grown in liquid Luria Broth (LB) or on LB agar plates with appropriate antibiotics at 37 °C. For HBA and HBAD production, 5 ml overnight cultures of recombinant *E. coli* strains were inoculated in 500 ml 2 × YT medium at 37 °C with shaking (200 r.p.m.). When the $OD_{600}$ of the cultures reached 0.8–1.0, IPTG was added to a final concentration of 0.4 mM. The cultures were then incubated at 28 °C for 24 h. To produce CBAD and other corrinoids, seed cultures of the recombinant strains were grown in LB medium at 37 °C overnight at 220 r.p.m. The seed cultures were used to inoculate 3 L shaker flask containing 600 ml TYG medium, a modified

medium[41] containing: $5\,g\,l^{-1}$ yeast extract, $10\,g\,l^{-1}$ tryptone, $5\,g\,l^{-1}$ $KH_2PO_4$, $2\,g\,l^{-1}$ glycine, $10\,g\,l^{-1}$ succinic acid, $20\,mg\,l^{-1}$ $CoCl_2\cdot6H_2O$, supplemented with $10\,g\,l^{-1}$ glucose and appropriate antibiotics at an inoculation volume of 5% at 37 °C in a rotary shaker (220 r.p.m) until $OD_{600}$ reached 0.8–1.0; 0.4 mM IPTG was then added to the cultures and the cultures were incubated at 28 °C for 24 h.

For the biosynthesis of APP, the overnight seed cultures were used to inoculate 50 ml of M9 mineral salt plus 0.5% yeast extract (M9Y medium) at an inoculation volume of 1% in 250 ml flasks at 37 °C until $OD_{600}$ reached 0.6–0.8. At this point, 0.4 mM IPTG and $5\,g\,l^{-1}$ L-threonine were added to the cultures, which were allowed to continue growth at 28 °C for 20 h. For de novo biosynthesis of vitamin $B_{12}$, strains were initially inoculated into LB medium at 37 °C (or 30 °C for recombinant strains harboring plasmids with the pSC101 replicon) overnight at 220 r.p.m. The seed cultures were used to inoculate 250 ml flasks containing 25 ml of CM medium containing $5\,g\,l^{-1}$ yeast extract, $10\,g\,l^{-1}$ tryptone, $5\,g\,l^{-1}$ $KH_2PO_4$, $10\,g\,l^{-1}$ glucose, $2\,g\,l^{-1}$ glycine, $10\,g\,l^{-1}$ succinic acid, $10\,g\,l^{-1}$ betaine, $20\,mg\,l^{-1}$ $CoCl_2\cdot6H_2O$, and $90\,mg\,l^{-1}$ 5,6-Dimethylbenzimidazole(DMBI), pH 6.5 with an starting $OD_{600}$ of 0.15. When the strains were grown at 30 °C and 220 r.p.m. to an $OD_{600}$ of 0.8–1.0, $50\,mg\,l^{-1}$ L-threonine and moderate IPTG was added to the cultures and the cultures were then incubated at 28 °C for 20 h.

Appropriate antibiotics were added to medium at the following concentrations: Kanamycin, $50\,\mu g\,ml^{-1}$; Ampicillin, $100\,\mu g\,ml^{-1}$; streptomycin, $50\,\mu g\,ml^{-1}$; Chloramphenicol, $34\,\mu g\,ml^{-1}$.

**Strain and plasmid construction**. *E. coli* DH5α was used as the host for cloning. *E. coli* BL21 (DE3) was used as the host for protein expression and purification. *E. coli* MG1655 (DE3) was used as the host for de novo production of vitamin $B_{12}$ and its intermediates. Plasmids pET28a, pCDFDuet-1, and pACYCDuet-1 were used for plasmid construction and protein expression. The details of the strains and plasmids used are described in Supplementary Data 1. The primers used are listed in Supplementary Data 2.

Construction of Module 1: All genes derived from *R. capsulatus*, *B. melitensis*, and *S. meliloti* were obtained by polymerase chain reaction (PCR) using genomic DNA templates. The HBA pathway was cloned from pET3a-cobAIGJMFKLH to pET28a by digestion with XbaI and BamHI, gel purification, and ligation, yielding pET28-HBA. A truncated *cobH* fragment with a C-terminal hexa-his tag and pET28-HBA were both digested by BmgBI and BamHI, purified and ligated with T4 DNA ligase, yielding pET28-HBA-his.

Construction of Module 2: The *cobB* gene was cloned to pCDFDuet-1 at BamHI and HindIII site, yielding pCDF-cobB-his. The other *cobB* gene was cloned to pCDFDuet-1 at NcoI site via golden gate[42] to create pCDF-cobB. The *cobN* genes from *B. melitensis*, *S. meliloti*, and *R. capsulatus* were cloned to pCDF-cobB via Gibson assembly[43], yielding pCDF-cobB-BmcobN, pCDF-cobB-SmcobN, pCDF-cobB-RccobN. The plasmids pCDF-cobB-BmcobN-his, pCDF-cobB-SmcobN-his, and pCDF-cobB-RccobN-his were constructed analogously, but with *cobN* with an N-terminal hexa-his tag instead. Amplified *cobS* and *cobT* fragments from *B. melitensis*, *S. meliloti*, and *R. capsulatus* were successively cloned to pACYCDuet-1 via Gibson assembly, yielding, respectively, pACYC-BmcobS-BmcobT, pACYC-SmcobS-SmcobT, and pACYC-RccobS-RccobT. To add a C-terminal hexa-his tag to *cobT* gene, inverse primers were designed to amplify the vectors. The resulting PCR products were phosphorylated by PNK and re-circularized, yielding pACYC-BmcobS-BmcobT-his, pACYC-SmcobS-SmcobT-his and pACYC-RccobS-RccobT-his. pACYC-BmcobS-his-BmcobT-his, pACYC-SmcobS-his-SmcobT-his, and pACYC-RccobS-his-RccobT-his were constructed analogously to add a C-terminal hexa-his tag to the *cobS* gene. The *BmcobST* pseudo-operon with promoters was amplified from pACYC-BmcobS-BmcobT and cloned into the XhoI site of pCDF-cobB-BmcobN and pCDF-cobB-BmcobN-his via Gibson assembly, yielding, respectively, pCDF-cobB-BmcobN-BmcobS-BmcobT and pCDF-cobB-BmcobN-his-BmcobS-BmcobT. Plasmids including pCDF-cobB-SmcobN-SmcobS-SmcobT, pCDF-cobB-SmcobN-his-SmcobS-SmcobT, pCDF-cobB-RccobN-RccobS-RccobT, pCDF-cobB-RccobN-his-RccobS-RccobT, pCDF-cobB-BmcobN-his-SmcobS-SmcobT, pCDF-cobB-BmcobN-his-RccobS-RccobT, pCDF-cobB-SmcobN-his-BmcobS-BmcobT, pCDF-cobB-SmcobN-his-RccobS-RccobT, pCDF-cobB-RccobN-his-BmcobS-BmcobT, and pCDF-cobB-RccobN-his-SmcobS-SmcobT were constructed analogously. The *cobW* genes from *B. melitensis*, *S. meliloti*, and *R. capsulatus* were cloned to pTrc99a at EcoRI and XbaI site, yielding, respectively, pTrc99a-BmcobW, pTrc99a-SmcobW, and pTrc99a-RccobW. The *cobW* gene from *B. melitensis* with its trc promoter was amplified from pTrc99a-BmcobW and then cloned to pCDF-cobB-BmcobN-his-BmcobS-BmcobT at XhoI site via Gibson assembly, yielding pCDF-cobBBmcobNSTW. Plasmids pCDF-cobBSmcobNSTW, pCDF-cobBRccobNSTW, pCDF-cobB-BmcobN-his-SmcobSTW, pCDF-cobB-BmcobN-his-RccobSTW, pCDF-cobB-SmcobN-his-BmcobSTW, pCDF-cobB-SmcobN-his-RccobSTW, pCDF-cobB-RccobN-his-BmcobSTW, and pCDF-cobB-RccobN-his-SmcobSTW were constructed analogously.

Construction of Module 3: A DNA fragment containing tac, lacUV5 promoters and a bidirectional terminator (Supplementary Fig. 18) was synthesized by GENEWIZ (China). This fragment was ligated via Gibson assembly into an amplified pACYCDuet-1 backbone lacking T7 promoters, multiple cloning sites, or terminators, yielding p15ASI. The *cbiMNQO* operon was amplified from *R. capsulatus* genome and cloned to p15ASI via Gibson assembly, yielding p15ASI-cbiMNQO.

Construction of Module 4: The codon-optimized gene cobR from B. melitensis was cloned into vector pACYCDuet-1 via Gibson assembly, yielding pACYC-BmcobR. The genes encoding cob(I)yrinic acid a,c-diamide adenosyltransferase and AdoCby synthase from E. coli, S. typhimurium, and B. melitensis were cloned into pACYC-BmcobR via Gibson assembly, yielding, respectively, pACYC-BmcobR-BtuRcbiP, pACYC-BmcobR-StcobAcbiP, and pACYC-BmcobROQ. The gene pduX from S. typhimurium and codon-optimized bluE from R. capsulatus were amplified and cloned into NcoI and XhoI sites, yielding pACYC-pduX and pACYC-bluE. The plasmids pACYC-his-pduX and pACYC-MBP-bluE were constructed via site directed mutagenesis[44]. The genes cobD from S. typhimurium and cobC from R. capsulatus were amplified and cloned into the NdeI sites of pACYC-his-pduX and pACYC-MBP-bluE, respectively, yielding pACYC-his-pduX-StcobD and pACYC-MBP-bluE-RccobC. The gene cbiB from S. typhimurium was cloned into vector pACYCduet-1 via Gibson assembly, yielding pACYC-cbiB. The cobD gene from R. capsulatus was cloned into vector pET28a, yielding pET28a-RccobD. The cobD expression cassette was cloned into pACYC-MBP-bluE-RccobC and pACYC-his-pduX-StcobD, respectively, yielding pACYC-MBP-bluE-RccobCD and pACYC-his-pduX-StcobD-RccobD. The cbiB expression cassette was cloned into pACYC-his-pduX-StcobD and pACYC-MBP-bluE-RccobC, respectively, yielding pACYC-his-pduX-StcobD-cbiB and pACYC-ECB. The pduX-StcobD-cbiB expression cassette was cloned into pACYC-BmcobR-BtuRcbiP, pACYC-BmcobR-StcobAcbiP, and pACYC-BmcobROQ, respectively, yielding pACYC-RRPXDB, pACYC-RAPXDB, and pACYC-BMXDB. The bluE-RccobCD expression cassette was cloned into pACYC-BmcobR-BtuRcbiP, pACYC-BmcobR-StcobAcbiP, and pACYC-BmcobROQ, yielding, respectively, pACYC-RRPECD, pACYC-RAPECD, and pACYC-BMECD. The bluE-RccobC-cbiB expression cassette was cloned into pACYC-BmcobROQ, pACYC-BmcobR-BtuRcbiP, and pACYC-BmcobR-StcobAcbiP, yielding, respectively, pACYC-BMECB, pACYC-RRPECB, and pACYC-RAPECB. Thr pduX-StcobD-RccobD expression cassette was cloned into pACYC-BmcobR-BtuRcbiP, pACYC-BmcobR-StcobAcbiP, and pACYC-BmcobROQ, yielding, respectively, pACYC-RRPXDD, pACYC-RAPXDD, and pACYC-BMXDD.

Construction of Module 5: Module 5 comprised four native genes from E. coli: cobU, cobT, cobS, and cobC.

Construction of Module 6: A fragment containing the Pr promoter, MicC Scaffold, B0015 double terminator, and BioBrick cloning sites was synthesized and cloned into pUC57 by GENEWIZ, yielding pKDG. This plasmid was used as the basis for gene inhibition via small RNA[31]. The plasmid pKDG was amplified using inverse primers containing respective around 24 bp N terminal sequences for the hemE, hemF, hemG, and hemH genes; amplified gene fragments were then phosphorylated by PNK and re-circularized, yielding plasmids pKDG-anti-hemE, pKDG-anti-hemF, pKDG-anti-hemG, and pKDG-anti-hemH. A fragment containing the hemF regulation module (around 300 bp from Pr promoter to B0015 double terminator) amplified from the pKDG-anti-hemF plasmid was digested by XbaI and PstI, and then purified and ligated with SpeI- and PstI-digested pKDG-anti-hemE, yielding pKDG-anti-hemEF. The other plasmids including pKDG-anti-hemEG, pKDG-anti-hemEH, pKDG-anti-hemFG, pKDG-anti-hemFH, and pKDG-anti-hemGH were constructed analogously. The hemO gene from R. palustris and the hemB, hemC, and hemD genes from S. meliloti were cloned together into p15ASI via Gibson assembly, yielding p15ASI-hemOBCD. The respective regulation modules of small RNA were amplified from pKDG-anti-hemE, pKDG-anti-hemF, pKDG-anti-hemG, pKDG-anti-hemH, pKDG-anti-hemEF, pKDG-anti-hemEG, pKDG-anti-hemEH, pKDG-anti-hemFG, pKDG-anti-hemFH, and pKDG-anti-hemGH were cloned into the BamHI site of pET28-HBA, yielding pET28-HBA-anti-hemE, pET28-HBA-anti-hemF, pET28-HBA-anti-hemG, pET28-HBA-anti-hemH, pET28-HBA-anti-hemEF, pET28-HBA-anti-hemEG, pET28-HBA-anti-hemEH, pET28-HBA-anti-hemFG, pET28-HBA-anti-hemFH, and pET28-HBA-anti-hemGH.

Construction of plasmids containing Modules 2 and 4 with different copy numbers: The plasmids pCDF-cobB-BmcobN-his-SmcobSTW and pACYC-RAPXDB were digested by both NheI and XbaI, resulting in the following products: the backbones of the two plasmids, the CloDF13 replicon, and the p15A replicon. The pSC101 replicon was obtained from the plasmid pKD46 by digestion with NheI and XbaI. The digested p15A and pSC101 replicons were each cloned into the NheI and XbaI sites of pCDF-cobB-BmcobN-his-SmcobSTW (backbone), generating the pCDF-Cby-p15A and pCDF-Cby-pSC101 plasmids, respectively. The digested CloDF13 and pSC101 replicons were cloned into the NheI and XbaI sites of pACYC-RAPXDB (backbone), generating the pACYC-Cbi-CloDF13 and pACYC-Cbi-pSC101 plasmids, respectively.

Construction of E. coli recombinant strains: The T7 RNA polymerase expression cassette was amplified from the E. coli BL21 (DE3) genome and integrated into the lacZ locus of E. coli MG1655 via λ/red recombination[45], resulting in MG1655 (DE3). Briefly, a kanamycin-sacB cassette with upstream and downstream homologous arms was transformed into E. coli MG1655 harboring pKD46. The recombinant strains were initially screened on LB agar plates containing 30 mg l[−1] kanamycin. The T7 RNA polymerase expression cassette with upstream and downstream homologous arms was transformed into the positive strain by electrotransformation. The second round of selection was carried out using sucrose resistance. Strains selected on LB agar plates containing 10% sucrose were confirmed by PCR, and then loss of pKD46 was induced by culturing at 37 °C. The genes ackA and pta were also deleted analogously. Module 3 and module 6

were integrated into the ldhA locus and the arabinose promoter locus, respectively, using CRISPR/Cas9[46]. The genes endA and gldA were also deleted using CRISPR/Cas9[46]. Briefly, the plasmids for gene knockout containing a 20-nucleotide target sequence–guide RNA scaffold hybrid (sequences listed in Supplementary Data 2), donor DNAs, the Cas9 nuclease and λ/red recombinases were constructed by Gibson assembly. Each plasmid was transformed into E. coli and plated onto LB plates supplemented with kanamycin and 1% (w/v) glucose. The resulting strains were inoculated into LB medium at 30 °C, 180 r.p.m. for 2 h, at which point the Cas9 nuclease was expressed by the addition of 10 mM L-arabinose. The cultures were incubated for an additional 4 h, and then plated onto LB plates supplemented with kanamycin and 10 mM L-arabinose. Mutants were identified by colony PCR and DNA sequencing. Finally, the mutants were incubated at 37 °C overnight to cure the temperature sensitive plasmids. The other recombinant strains were constructed by transformation of corresponding plasmids (listed in Supplementary Data 1).

**Plasmid stability assays.** At the 10 h and 20 h time points after IPTG induction, 100 µl samples of the E. coli recombinant strains were taken from the induced cultures and adjusted to $OD_{600}$ of 1.0 in fresh LB medium, and serial dilutions of these adjusted sample cultures were plated onto non-selective LB and LB agar plates with kanamycin, streptomycin, or chloramphenicol, and incubated at 37 °C (or 30 °C for plasmids with the thermo-sensitive pSC101 replicon) until colonies were visible. The number of colonies on each plate were scored and used to calculate viable counts and individual plasmid stabilities.

**Corrinoids biosynthesis and purification.** For routine separation and purification, corrinoids were extracted from recombinant E. coli strains harboring corresponding plasmids via ion exchange chromatography. Briefly, cells were harvested by centrifugation at 5000 × g for 20 min and re-suspended in IEX-buffer A containing 20 mM Tris-HCl (pH 7.4) and 100 mM NaCl. The cell suspension was disrupted twice using a JN-3000 Plus homogenizer at 1200 v and centrifuged at 11,000 × g for 1 h. The supernatant was passed through a DEAE-Sephadex A25 column. The column was washed with five column volumes of IEX-buffer B containing 20 mM Tris-HCl (pH 7.4) and 300 mM NaCl. Corrinoids were eluted with IEX-buffer C containing 20 mM Tris-HCl (pH 7.4) and 2 M NaCl. Samples were then concentrated and filtered using 0.22 µm filters prior to HPLC analysis.

HBA and HBAD were extracted via a modified enzyme-trap method[47]. Briefly, cells were resuspended in enzyme-trap buffer A (100 mM NaCl, 5 mM imidazole, 20 mM HEPES, pH 7.5) and disrupted twice using a JN-3000 Plus homogenizer at 1200 v and centrifuged at 11,000 × g for 1 h. The supernatant was filtered using a 0.22 µm filter. The filtrate was then loaded onto an equilibrated Ni Sepharose column (GE Healthcare). After washing with 10 column volumes of enzyme-trap buffer B (100 mM NaCl, 60 mM imidazole, 20 mM HEPES, pH 7.5), the proteins were eluted using enzyme-trap buffer C (100 mM NaCl, 400 mM imidazole, 20 mM HEPES, pH 7.5).

**Gene expression and protein production and purification.** Purified proteins were used for enzyme assays. For the production of recombinant proteins, E. coli BL21 (DE3) cells carrying recombinant plasmids were pre-inoculated into 5 ml LB liquid media and overnight cultures were inoculated into 500 ml of fresh LB media containing appropriate antibiotics. The cultures were left to grow at 37 °C until the $OD_{600}$ reached 0.6 and were then induced with 0.5 mM IPTG at 30 °C for 12 h. Protein purification was performed by metal chelated affinity chromatography using a Ni Sepharose column (GE Healthcare). Briefly, cells were harvested by centrifugation at 5000 × g for 20 min and re-suspended in buffer A (20 mM sodium dihydrogen phosphate, 500 mM sodium chloride, 30 mM imidazole, pH 7.4). The cell suspension was disrupted using a JN-3000 Plus homogenizer at 1200 v and centrifuged at 11,000 × g for 1 h. The supernatant was filtered using a 0.22 µm filter. The filtrate was then loaded onto an equilibrated Ni Sepharose column. After washing three times with 10 column volumes of buffer B (20 mM sodium dihydrogen phosphate, 500 mM sodium chloride, 100 mM imidazole, pH 7.4), the proteins were eluted using buffer C (20 mM sodium dihydrogen phosphate, 500 mM sodium chloride, 500 mM imidazole, pH 7.4). The protein storage buffer was then exchanged for buffer I (50 mM Tris-HCl, pH 7.5, 50 mM sodium chloride, 50% (v/v) glycerol) through ultrafiltration using Millipore's Amicon® Ultra-15 centrifugal filters and stored at −20 °C. The protein content of the samples was analyzed by SDS-PAGE. Proteins were quantified using a Bradford Protein Assay Kit (Beijing Solarbio Science & Technology Co., Ltd) using bovine serum albumin as a standard.

**In vitro enzyme assay of CobB.** CobB was assayed as reported previously[9]. Briefly, 5 µM HBA, 0.1 M Tris hydrochloride (pH 7.6), 1 mM ATP, 2.5 mM magnesium chloride, and 1 mM glutamine in a total volume of 250 µl were combined and incubated at 30 °C for 60 min. The reaction was terminated by addition of 250 µl of 1 M hydrochloric acid. HBAD production was monitored with LC-MS (see Analytical methods below).

**In vitro enzyme assay of the CobNST proteins.** The CobNST proteins were assayed by adapting a previously described method[21]. The reaction mixture

consisted of 0.1 M Tris hydrochloride (pH 8.0), 100 μM cobalt chloride, 7.5 mM ATP, 6.5 mM magnesium chloride, and purified CobST in a $12.4 \times 12.4 \times 45$ mm quartz micro-cuvette. The reaction was initiated by addition of CobN-HBAD complex purified using the enzyme-trap method (above), and absorbance was measured once each min for 20 min at 305–600 nm at 30 °C. The control lacked cobalt chloride in the reaction mixture. Cobyrinic acid a,c-diamide production was monitored using LC-MS (see Analytical methods below).

**In vitro enzyme assay of BluE**. Standard assay mixtures for BluE were modified from a previous report[14]: 15 mM HEPES, pH 7.4, 20 mM NaCl, 10 mM MgCl₂, 200 μM ATP, and 500 μM L-threonine were combined in a total volume of 200 μl. The reaction was initiated by the addition of 10 μg crude BluE. The reaction was stopped by incubation at 100 °C for 10 min, and L-threonine phosphate was monitored using LC-MS (see Analytical methods below).

**In vitro enzyme assay of RcCobC**. The L-threonine-O-3-phosphate decarboxylase assay was conducted based on previous study[13]. Specifically, reactions had a 100 μl final volume and contained 50 mM PIPES, pH 6.8, 500 μM Thr-P, 500 μM pyridoxal phosphate; the reaction was initiated by the addition of 100 μg purified RcCobC. After incubation at 37 °C for 90 min, 178 μL of the reaction mixture was diluted to 200 μl with 20 μl NEB CutSmart buffer and 2 μl calf intestinal alkaline phosphatase (CIP). The mixture was incubated at 37 °C for 30 min, and the reaction was subsequently stopped by incubation at 100 °C for 10 min. After centrifugation at 10,000 × g for 2 min, AP in the supernatant was detected by HPLC (see Analytical methods below).

**Analytical methods**. HBA, HBAD, and CBAD were measured using an Agilent 1260 High Performance Liquid Chromatography (HPLC) system equipped with a diode array detector and a SB-Aq column (4.6 × 150 mm, 5 μm, Agilent); samples were analyzed at 30 °C and signals were monitored at 329 nm. Samples containing HBA, HBAD, and CBAD were filtered using a 0.22 μm filter, and a 20 μl portion was directed analyzed via HPLC. The mobile phase was water (solvent A) and methanol (solvent B) (both contain 0.1% formic acid) at a flow rate of 1 ml/min. Elution gradients were: 0–25% B (0–2 min), 25–34% B (2–4 min), 34–70% B (4–12 min), 70–100% B (12–17 min), 100% B (17–23 min), 0% B (23–25 min), and 0% B (25–32 min). The molar extinction coefficient of HBAD is available. To determine the HBA concentration of extracted HBA, it was completely converted to HBAD by CobB enzyme from R. capsulatus in vitro. Then HBA attained by enzyme-trap method was used as a reference to quantify HBA production of recombinant strains. HBA in samples were identified and quantified by comparison to standard HBA attained from FH-HBA, a strain harboring pET28-HBA-his (Supplementary Data 1) via an enzyme-trap method, with some modifications[47]. Corrinoids were extracted for LC-MS analysis from 600 mL of cell culture via ion exchange chromatography, and 35 μl of concentrated sample was injected for analysis. The reaction mixtures from in vitro assays were filtered using 0.22 μm filters and then directly injected for analysis. LC-MS analysis was carried out using an Agilent 1260 HPLC equipped with a SB-Aq column (4.6 × 150 mm, 5 μm, Agilent) and a Bruker microQ-TOF II mass spectrometer equipped with an ESI ionization source.

L-threonine, L-threonine phosphate and AP in the supernatant of the fermentation culture were detected by HPLC (Agilent 1260) with a Zorbax Eclipse-AAA column (4.6 × 150 mm, 5 μm particle size, Agilent) at 40 °C monitored at 338 nm after automated online derivatization using o-phthalaldehyde. The elution was performed using a gradient of solvent A (40 mM NaH₂PO₄•2H₂O, pH 7.8) and solvent B (methanol/acetonitrile/water = 45:45:10, by vol.) at 2 ml min⁻¹. Elution gradients were: 0 to 1.9 min, 0% B; 18.1 min, 57% B; 18.6 min, 100% B; 22.3 min, 100% B; 23.2 min, 0% B; 26 min, 0% B. Twenty microliters of the in vitro enzyme assay product of BluE was injected for LC-MS analysis. LC-MS analysis was carried out using an Agilent 1260 HPLC equipped with a Zorbax Eclipse-AAA column (4.6 × 150 mm, 5 μm particle size, Agilent) coupled to a Bruker microQ-TOF II mass spectrometer equipped with an ESI ionization source. The flow rate was reduced to 1 ml/min during LC-MS analysis, while other conditions remain unchanged.

Vitamin B₁₂ was prepared as follows: The bacterial pellet from 20 ml fermentation broth was collected and resuspended with ddH₂O to a final volume of 1.3 ml; and then 130 μl of NaNO₂ 8% (w v⁻¹), 130 μL of glacial acetic acid and 10 μl of NaCN 1% (w v⁻¹) were added; thereafter, the mixture was boiled for 30 min and centrifugated at 13,000 × g for 10 min. The supernatant was resolved on a reverse phase C-18 column (4.6 × 250 mm, 5 μm, Agilent) by HPLC (Agilent 1260) operating at 30 °C monitored at 361 nm. The mobile phase consisted of 30% methanol at a flow rate of 0.8 ml min⁻¹ for 15 min.

## Data availability

All data generated in the present study and included in this article and its supplementary information files are available from the corresponding author upon reasonable request.

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

## Acknowledgements

We thank D. K. Newman (California Institute of Technology) for the gift of the strain *Rhodobacter capsulaus* SB1003, Qingmin Wu (China Agricultural University) for the gift of the strain *Brucella melitensis* bv. 1 str. 16 M, and Martin J. Warren (University of Kent) for the gift of pET3a-cobAIGJMFKLH. This work was supported by the Tianjin Science Fund for Distinguished Young Scholars (17JCJQJC45300), the Tianjin Science and Technology Project (15PTCYSY00020), the Key Projects of the Tianjin Science & Technology Pillar Program (14ZCZDSY00058), the Science and Technology Service Network (STS) Initiative of Chinese Academy of Sciences (CAS), and the Key Laboratory of Systems Microbial Biotechnology, Tianjin Institute of Industrial Biotechnology of the Chinese Academy of Sciences.

## Author contributions

H.F. and D.Z. conceived and designed the experiments; H.F., D.L., J.K., and P.J. performed the experiments; H.F. and D.Z. analysed the experimental data; H.F., J.S., and D. Z. prepared the manuscript.

## Additional information

**Competing interests:** H.F. and D.Z. declare the following competing interests: this work described here has been included in 2 Chinese patent applications (201711296890.5 and 201711296896.2) by the Tianjin Institute of Industrial Biotechnology of the Chinese Academy of Sciences. The remaining authors declare no competing interests.

