## [Peer Review File · Nature Communications]

Reviewers' Comments:

Reviewer #1:

Remarks to the Author:

Fang et al reported the biosynthesis of vitamin B12 using *E. coli* as the host strain. The synthetic pathway involving over 30 genes was divided into 6 modules to optimize the productivity of vitamin B12. The modular optimization was successfully demonstrated, and the amount of vitamin B12 produced by the engineered strain was over 200-fold higher than the primary strain. In addition, they identified that the aerobic pathway to synthesize vitamin B12 in *Rhodobacter capsulatus* is the same as the anaerobic pathway distributed in other species. This is a well-organized research with high significance. However, several points have to be revised and clarified before publication.

The amount of vitamin B12 produced by using the optimized strain reached 307.00 micro-g/g DCW, which was over 200-fold higher than the primary strain. The productivity was drastically increased. To emphasize the achievement, the productivity of vitamin B12 in other host strains, such as one used for the industrial production, should be referred in introduction or discussion section.

In figure 5, a large number of transformants created in this study are referred. It is difficult for many readers to follow the phenotype of each strain (eg. what strains express *cobW* gene? what difference are there between figure ab, cd, and ef? More visual explanation is significant in figures including a lot of phenotypes.). Figure 5 should be revised as easy-to-follow. It is helpful for many readers.

In final section in results, culture medium and condition were optimized. Although some expensive reagents were added to culture medium, Is this result be applicable to the industrial production of vitamin B12? Also, is the addition of succinic acid for cell growth? What for were glycine and betaine added?

Vitamin B12 is composed of 63 numbers of carbon molecules. Some researchers in metabolic engineering area are really interested in the origin of carbon molecules in a target compound. What from are the carbons in vitamin B12 produced in this study derived? From glucose, amino acids added into medium, tryptone.....and so on. Do you have any ideas or data?

Minor points

line 138 In supplementary figure 1(a), in the case of strain expressing *CobB* (blue), two peaks seems to be confirmed. Is there a possibility that HBA was converted to other compounds except for HBAD by *CobB*?

line 138 In supplementary figure 1(b), please show the results of LC/MS analysis of standard samples of HBA and HBAD.

line 167 Are there any differences of LC chromatograms between HBAD and CBAD? In supplementary figure 4 or 5, is the LC chromatogram of CBAD in LC-MS analysis available?

Where does supplementary figure 6 refer?

line 206,207 "(R)-1-Amino-2-propanol" change to "AP" Please use abbreviations.

line 226 In supplementary figure 9, please confirm Thr-P formation by MS analysis.

line 337 "Fig 5c" change to "Fig 5c, d"

line 455 Please include some references about vitamin B12 production in *P. denitrifians*.

Figure 1 “uroporphyrinogen III” change to “uroporphyrinogen III (Uro III),” “adenosylcobinamide” change to “adenosylcobinamide (AdoCbi)” and so on. Please include abbreviations.

Reviewer #2:

Remarks to the Author:

This paper is described construction and production of vitamin B12 by genetically engineered E. coli. Although genetically engineered E. coli is quite popular now, vitamin B12 biosynthesis is very complicate and difficult to engineer the producer strains; authors in this paper successfully overproduced it by engineered E. coli by means of heterologous expression of a total of 28 genes from various bacteria with their tremendous efforts. They increased the yield of vitamin B12 using the recombinant E. coli strain by more than ~250-fold or 307 µg/g DCW via metabolic engineering and optimization of fermentation conditions. However the 250-fold is no point because the basis level of B12 in the host strain of E. coli is quite low. The paper also clarified that the biosynthetic steps from co (II) byrinic acid a,c-diamide to adocobalamin are the same in both the aerobic and anaerobic pathways since this pathway was unknown until now. Analyses of products in each step or module are well done. Although most of the genes are not new or used already developed techniques, this study will contribute to the field of genetically engineering as an example of how around 30 enzymes concerned with a complex biosynthetic pathway can be transferred between organisms to facilitate practical production. In this meaning, this paper has an impact to persuade a challenge in basic and applied sciences.

Comments:

1. Line 57: At the introduction, authors mentioned that “the industrial production of vitamin B12 is mainly from fermentation of *Pseudomonas denitrifians* and *Propionibacterium shermanii*, but these strains grow slow and are difficult to engineer”. The slow growing of these strains may be true compared with E. coli, but there are many reports of genetic trials in both bacteria and thus authors have to show fairly these several papers reported about genetic improvement of B12 production in both industrial stains.
2. As the references for ALA, “Biotechnology & Genetic Engineering Reviews. Vol. 18” (ed. S. E. Harding), pp. 149-170, Intercept, Andover, England (2001) and “Plant Adaptation to Environmental Change: Significance of Amino Acids and their Derivatives” (eds. NA Anjum, SS. Gill &R. Gill), pp.18-34, CABI, Oxford shire, UK (2014); as for vitamin B12, Le Lait, 85, 9-22 (2005) are good review papers to be cited.
3. In 6 modules constructed, are there any incompatibilities of plasmids in the host bacterium? The paper should be mentioned the stability of each plasmid (module) during fermentation.
4. The effect of copy numbers of plasmid modules in E. coli on production of vitamin B12 should be discussed.
5. The final yield of vitamin B12 by the engineered E. coli should be evaluated in the comparison with those of *Pseudomonas denitrifians* and *Propionibacterium shermanii*.
6. Have you ever tried to change the codon usage of the heterologous bacteria to that of E. coli since some genes from Gram positive bacteria must be high GC contents?
7. Did you find any feedback control of enzymes by accumulation of intermediates or final products? This information is valuable in porphyrin biosynthesis.

We would like to thank both you and the reviewers for your time and effort spent on our behalf in the evaluation of our manuscript. The guidance and reviewer comments and valuable suggestions have helped to clarify many important points and have improved the quality of our study substantially. We have revised the manuscript according to the reviewers' suggestions, and we have detailed the changes and corrections in our point-by-point response, below.

Reviewer #1 (Remarks to the Author):

Fang et al reported the biosynthesis of vitamin B12 using *E. coli* as the host strain. The synthetic pathway involving over 30 genes was divided into 6 modules to optimize the productivity of vitamin B12. The modular optimization was successfully demonstrated, and the amount of vitamin B12 produced by the engineered strain was over 200-fold higher than the primary strain. In addition, they identified that the aerobic pathway to synthesize vitamin B12 in *Rhodobacter capsulatus* is the same as the anaerobic pathway distributed in other species. This is a well-organized research with high significance. However, several points have to be revised and clarified before publication.

Comments:

The amount of vitamin B12 produced by using the optimized strain reached 307.00 micro-g/g DCW, which was over 200-fold higher than the primary strain. The productivity was drastically increased. To emphasize the achievement, the productivity of vitamin B12 in other host strains, such as one used for the industrial production, should be referred in introduction or discussion section.

RESPONSE: First, let us thank you for your careful review of our work and your insightful and helpful suggestions to improve our study. Regarding this comment specifically, the productivity of vitamin B₁₂ in other host strains including *Bacillus megaterium* (Biedendieck, Malten et al. 2010) and the industrial strains *Pseudomonas denitrifians* (Li, Liu et al. 2008) and *Propionibacterium freudenreichii* (VY, NI et al. 1998) were compared with the optimized *E. coli* strain in the discussion section of the revised manuscript.

COMMENT: In figure 5, a large number of transformants created in this study are referred. It is difficult for many readers to follow the phenotype of each strain (eg. what strains express *cobW* gene? what difference are there between figure ab, cd, and ef? More visual explanation is significant in figures including a lot of phenotypes.). Figure 5 should be revised as easy-to-follow. It is helpful for many readers.

RESPONSE: Thank you for this suggestion. We have now added information for the key phenotypes of the strains in Fig. 5 to make it easily understandable and easier to

follow.

COMMENT: In final section in results, culture medium and condition were optimized. Although some expensive reagents were added to culture medium, Is this result be applicable to the industrial production of vitamin B12? Also, is the addition of succinic acid for cell growth? What for were glycine and betaine added?

RESPONSE: The reagents used in this study such as glycine, succinic acid, and betaine are common supplements for industrial utilization, and these supplements are much cheaper than vitamin B12 (4 \$/kg for glycine, 4 \$/kg for succinic acid, 2 \$/kg for betaine and 1628 \$/kg for vitamin B12). Succinyl-CoA and glycine are precursors of ALA (δ -aminolevulinate) through C₄ pathway. Succinic acid can be transformed to succinyl-CoA by succinyl-CoA synthetase in the TCA cycle. These two compounds are usually added to the culture medium for industrial production of ALA (Nishikawa and Murooka 2001).

Betaine, as a small zwitterionic compound is known to play critical roles in osmoregulation and in methionine synthesis (methyl-group donor) (L N Csonka and Hanson 1991, Kim, Choi et al. 2003). In *P. denitrificans*, methionine, one of the precursors of vitamin B₁₂, could be synthesized from homocysteine with betaine as the methyl group donor (Kim, Choi et al. 2003). Some reagents such as yeast extract, tryptone can be changed to the industrial grade, which are much cheaper and will be applicable for the industrial production of vitamin B₁₂.

COMMENT: Vitamin B12 is composed of 63 numbers of carbon molecules. Some researchers in metabolic engineering area are really interested in the origin of carbon molecules in a target compound. What from are the carbons in vitamin B12 produced in this study derived? From glucose, amino acids added into medium, tryptone.....and so on. Do you have any ideas or data?

RESPONSE: In terms of the structure of vitamin B₁₂, 37 carbons of which are derived from ALA; 8 carbons from SAM, 3 carbons from L-threonine, 5 carbons from nicotinic acid mononucleotide (NaMN) and 9 carbons from 5,6-Dimethylbenzimidazole (DMBI). ALA is derived from glucose or succinyl-CoA and glycine, so most of the carbons of vitamin B₁₂ are derived from glucose, DMBI, succinyl-CoA, and glycine. Owing to the complex metabolic pathways of these precursors and exogenous precursors supplemented in the medium, it is difficult to monitor the precise ratio of these precursors.

Minor points

COMMENT: line 138 In supplementary figure 1(a), in the case of strain expressing CobB (blue), two peaks seems to be confirmed. Is there a possibility that HBA was converted to other compounds except for HBAD by CobB?

RESPONSE: HBA is transformed to HBAD by two amidations, at positions a and c

carboxyl groups of HBA (Debussche, Thibaut et al. 1990). The first amino group is introduced at position c to form hydrogenobyric acid c-monoamide, which is further amidated at position a to produce HBAD (Galperin and Grishin 2000). In this *in vitro* assay, we detected both HBAD and the intermediate hydrogenobyric acid c-monoamide. The first and second peaks in supplementary figure 1(a) are HBA and hydrogenobyric acid c-monoamide, respectively, which were identified by MS analysis. This content is now presented in revised supplementary Figure 1a, b.

COMMENT: line 138 In supplementary figure 1(b), please show the results of LC/MS analysis of standard samples of HBA and HBAD.

RESPONSE: Standard samples of HBA and HBAD are shown in the results of LC/MS analysis in supplementary Figure 1b.

COMMENT: line 167 Are there any differences of LC chromatograms between HBAD and CBAD? In supplementary figure 4 or 5, is the LC chromatogram of CBAD in LC-MS analysis available?

RESPONSE: Due to the similar structure of HBAD and CBAD, it was difficult to separate them by LC chromatograms. We have successfully separated them by using a new HPLC method which has been updated in the revised manuscript. The peaks of HBAD and CBAD were identified by LC-MS analysis; this information has been added in the supplementary Fig 4 and 5 in the revised manuscript.

COMMENT: Where does supplementary figure 6 refer?

RESPONSE: Supplementary Figure 6 refer to Supplementary Note 1 in the Supplementary information. Now we have also added its reference in the revised manuscript.

COMMENT: line 206,207 “(R)-1-Amino-2-propanol” change to “AP” Please use abbreviations.

RESPONSE: Thanks for catching this; we have changed it in the revised manuscript.

COMMENT: line 226 In supplementary figure 9, please confirm Thr-P formation by MS analysis.

RESPONSE: We have now confirmed Thr-P formation via MS analysis; kindly see supplementary Figure 9.

COMMENT: line 337 “Fig 5c” change to “Fig 5c, d”.

RESPONSE: We have revised Fig. 5 and corrected references in the revised

manuscript accordingly.

COMMENT: line 455 Please include some references about vitamin B12 production in *P. denitrificans*.

RESPONSE: Thank you for this suggestion; we have now included some references about vitamin B₁₂ production in *P. denitrificans* in the discussion section in the revised manuscript. The references are shown as follows:

Li KT, Liu DH, Chu J, Wang YH, Zhuang YP, Zhang SL. An effective and simplified pH-stat control strategy for the industrial fermentation of vitamin B₁₂ by *Pseudomonas denitrificans*. *Bioprocess Biosyst Eng* **31**, 605-610 (2008).

Xia W, Chen W, Peng WF, Li KT. Industrial vitamin B₁₂ production by *Pseudomonas denitrificans* using maltose syrup and corn steep liquor as the cost-effective fermentation substrates. *Bioprocess Biosyst Eng* **38**, 1065-1073 (2015).

Li KT, *et al.* Improved large-scale production of vitamin B₁₂ by *Pseudomonas denitrificans* with betaine feeding. *Bioresource technology* **99**, 8516-8520 (2008).

COMMENT: Figure 1 “uroporphyrinogen III” change to “uroporphyrinogen III (Uro III),” “adenosylcobinamide” change to “adenosylcobinamide (AdoCbi)” and so on. Please include abbreviations.

RESPONSE: Thank you for your suggestion. We have corrected it.

Reviewer #2 (Remarks to the Author):

This paper is described construction and production of vitamin B12 by genetically engineered *E. coli*. Although genetically engineered *E. coli* is quite popular now, vitamin B12 biosynthesis is very complicate and difficult to engineer the producer strains; authors in this paper successfully overproduced it by engineered *E. coli* by means of heterologous expression of a total of 28 genes from various bacteria with their tremendous efforts. They increased the yield of vitamin B12 using the recombinant *E. coli* strain by more than ~250-fold or 307 µg/g DCW via metabolic engineering and optimization of fermentation conditions. However the 250-fold is no point because the basis level of B12 in the host strain of *E. coli* is quite low. The paper also clarified that the biosynthetic steps from co (II) byrnic acid a,c-diamide to adocobalamin are the same in both the aerobic and anaerobic pathways since this pathway was unknown until now. Analyses of products in each step or module are well

done. Although most of the genes are not new or used already developed techniques, this study will contribute to the field of genetically engineering as an example of how around 30 enzymes concerned with a complex biosynthetic pathway can be

transferred between organisms to facilitate practical production. In this meaning, this paper has an impact to persuade a challenge in basic and applied sciences.

Comments:

COMMENT: 1. Line 57: At the introduction, authors mentioned that “the industrial production of vitamin B12 is mainly from fermentation of *Pseudomonas denitrifians* and *Propionibacterium shermanii*, but these strains grow slow and are difficult to engineer”. The slow growing of these strains may be true compared with *E. coli*, but there are many reports of genetic trials in both bacteria and thus authors have to show fairly these several papers reported about genetic improvement of B12 production in both industrial stains.

RESPONSE: We would first like to thank the reviewer for their comments and helpful suggestions. In the revised manuscript, we now reference several representative papers about the engineering of *Pseudomonas denitrifians* and *Propionibacterium* strains in the introduction section. The references are shown as follows:

F B, B C, J C, L D, S L-S, D T. Polypeptides involved in the biosynthesis of cobalamines and/or cobamides, DNA sequences coding for these polypeptides, and their preparation and use. Europe Patent EP0516647B1 (1998).

Piao Y, Yamashita M, Kawaraichi N, Asegawa R, Ono H, Murooka Y. Production of vitamin B₁₂ in genetically engineered *Propionibacterium freudenreichii*. *J Biosci Bioeng* **98**, (2004).

Kiatpapan P, Murooka Y. Construction of an expression vector for propionibacteria and its use in production of 5-aminolevulinic acid by *Propionibacterium freudenreichii*. *Applied microbiology and biotechnology* **56**, 144-149 (2001).

Kiatpapan P, *et al.* Characterization of pRGO1, a Plasmid from *Propionibacterium acidipropionici*, and Its Use for Development of a Host-Vector System in Propionibacteria. *Applied and environmental microbiology* **66**, 4688-4695 (2000).

COMMENT: 2. As the references for ALA, “Biotechnology & Genetic Engineering Reviews. Vol. 18” (ed. S. E. Harding), pp. 149-170, Intercept, Andover, England (2001) and “Plant Adaptation to Environmental Change: Significance of Amino Acids and their Derivatives” (eds. NA Anjum, SS. Gill & R. Gill), pp.18-34, CABI, Oxford shire, UK (2014); as for vitamin B12, *Le Lait*, 85, 9-22 (2005) are good review papers to be cited.

RESPONSE: Thank you for your suggestion. We now refer to these references in the revised manuscript.

COMMENT: 3. In 6 modules constructed, are there any incompatibilities of plasmids in the host bacterium? The paper should be mentioned the stability of each plasmid (module) during fermentation.

RESPONSE: The plasmids containing Module 1, 2, and 4 have the pMB1, CloDF13, and p15A replicons, respectively. These plasmids are compatible. They are very stable when cultured without IPTG induction during fermentation. However, some plasmids become unstable after IPTG induction. In the case of the strain FH364, the plasmid containing Module 1 was very stable, even when cultured 20 h after IPTG induction. However, the plasmid maintenance ratio of the plasmids containing Module 2 and 4 were 52.66% and 63.94%, respectively. Please see revised Supplementary Table 2 for more information about plasmid stability tests for other strains. We can now conclude that it is high metabolic burden, but not the plasmid itself, that lead to plasmid instability, so we will use other methods such as expression of essential genes rather than using antibiotics to decrease plasmid instability in our future work. All of this information including Supplementary Table 1 and 2 has been added in the revised manuscript and revised supplementary data.

COMMENT: 4. The effect of copy numbers of plasmid modules in *E. coli* on production of vitamin B₁₂ should be discussed.

RESPONSE: We also investigated whether regulating copy numbers of plasmids could improve vitamin B₁₂ production further. The plasmids harboring Module 1, 2, and 4 contained the respective replicons pMB1 (medium copy numbers), CloDF13 (high copy numbers), and p15A (medium copy numbers). As the amount of HBA is sufficient for these recombinant *E. coli* strains, the replicon of this plasmid was not changed, while replicons of plasmids harboring Module 2 and 4 were changed, resulting in 5 strains harboring compatible plasmids with different copy numbers (Supplementary Table 1). We observed a decrease in vitamin B₁₂ production when the replicon of the plasmid harboring Module 3 was changed to the low copy number pSC101 (139.56 µg/g DCW vs. 171.81 µg/g DCW for FH364). In addition, when replicons of the plasmid harboring Module 2 were changed to the medium copy p15A or low copy pSC101, vitamin B₁₂ was not detected in the resulting four strains. These findings indicate that Module 2 can be a bottleneck if medium or low copy plasmids are used. Finally, these results also indicate the top performing strains should have plasmids containing Module 1, 2, and 4 with medium, high, and medium copy numbers, respectively. We have added these information in the results section of the revised manuscript.

COMMENT: 5. The final yield of vitamin B₁₂ by the engineered *E. coli* should be evaluated in the comparison with those of *Pseudomonas denitrifians* and *Propionibacterium shermanii*.

RESPONSE: Thank you for your suggestion. We have now included references and have compared vitamin B₁₂ production of *E. coli* with that of *Pseudomonas denitrifians* and *Propionibacterium freudenreichii* in the revised manuscript. The references are as follows:

Li KT, Liu DH, Chu J, Wang YH, Zhuang YP, Zhang SL. An effective and simplified pH-stat control strategy for the industrial fermentation of vitamin B₁₂ by *Pseudomonas denitrificans*. *Bioprocess Biosyst Eng* **31**, 605-610 (2008).

VY B, NI Z, AA E. Tetrapyrroles: diversity, biosynthesis, and biotechnology (review). *Applied Biochemistry and Microbiology* **34**, 1-18 (1998)

Biedendieck R, *et al.* Metabolic engineering of cobalamin (vitamin B₁₂) production in *Bacillus megaterium*. *Microb Biotechnol* **3**, 24-37 (2010).

Ko Y, *et al.* Coenzyme B₁₂ can be produced by engineered *Escherichia coli* under both anaerobic and aerobic conditions. *Biotechnology journal* **9**, 1526-1535 (2014).

COMMENT: 6. Have you ever tried to change the codon usage of the heterologous bacteria to that of *E. coli* since some genes from Gram positive bacteria must be high GC contents?

RESPONSE: All the heterologous bacteria we used here including *Rhodobacter capsulatus*, *Brucella melitensis*, *Sinorhizobium meliloti* 320, *Salmonella typhimurium*, and *Rhodopseudomonas palustris* are all Gram-negative bacteria. Only codons of *cobR* and *bluE* from *Brucella melitensis* were optimized, because we did not have the genome template at that time. To solve the problem of low expression of some heterologous proteins, we found that N terminal fusion hexa-histidine tags (“Biosynthesis of CBAD” section) or optimized 5’UTR (Supplementary Figure 8) were very useful.

COMMENT: 7. Did you find any feedback control of enzymes by accumulation of intermediates or final products? This information is valuable in porphyrin biosynthesis.

RESPONSE: We did not observe significant accumulation of intermediates or final products (none were accumulated to more than 1 mg/l); we did note that HBAD accumulated to 0.52 mg/l, owing perhaps to possible inefficiencies in the activities or expression levels of downstream enzymes; this is not yet clear. Nevertheless, it does not appear at this time that this pathway is feedback inhibited in our engineered system.

Nat Comm manuscript checklist,

Abstract: no more than 150 words

RESPONSE: We have revised the abstract and deleted several sentences.

Main text: no more than 5,000 words in total (Introduction, Results, Discussion)

RESPONSE: We have revised the main text and deleted some sentences.

Introduction: less than 1,000 words

RESPONSE: We have revised the introduction and deleted some sentences.

Figures: Avoid the use of red and green in figures to avoid confusion for colour-blind readers (magenta and turquoise are alternatives)

RESPONSE: We have changed red and green in figures to avoid confusion for colour-blind readers or red and green do not affect the audience reading and understanding.

We would again like to thank the reviewers for their very helpful and insightful review of our manuscript. We think the revised manuscript is improved, and hope that it is suitable for publication in Nature Communications.

Sincerely,

Dawei Zhang

REFERENCES

- Biedendieck, R., M. Malten, H. Barg, B. Bunk, J. H. Martens, E. Deery, H. Leech, M. J. Warren and D. Jahn (2010). "Metabolic engineering of cobalamin (vitamin B₁₂) production in *Bacillus megaterium*." Microb Biotechnol **3**(1): 24-37.
- Debussche, L., D. Thibaut, B. Cameron, J. Crouzet and F. Blanche (1990). "Purification and characterization of cobyrinic acid a,c-diamide synthase from *Pseudomonas denitrificans*." Journal of Bacteriology **172**(11): 6239-6244.
- Galperin, M. Y. and N. V. Grishin (2000). "The synthetase domains of cobalamin biosynthesis amidotransferases cobB and cobQ belong to a new family of ATP-dependent amidoligases, related to dethiobiotin synthetase." Proteins: Structure, Function, and Bioinformatics **41**(2): 238-247.
- Kim, S. K., K. H. Choi and Y. C. Kim (2003). "Effect of acute betaine administration on hepatic metabolism of S-amino acids in rats and mice." Biochemical Pharmacology **65**(9): 1565-1574.
- L N Csonka, a. and A. D. Hanson (1991). "Prokaryotic Osmoregulation: Genetics and Physiology." Annual Review of Microbiology **45**(1): 569-606.
- Li, K. T., D. H. Liu, J. Chu, Y. H. Wang, Y. P. Zhuang and S. L. Zhang (2008). "An effective and simplified pH-stat control strategy for the industrial fermentation of vitamin B₁₂ by *Pseudomonas denitrificans*." Bioprocess Biosyst Eng **31**(6): 605-610.
- Nishikawa, S. and Y. Murooka (2001). "5-Aminolevulinic acid: production by fermentation, and agricultural and biomedical applications." Biotechnology & genetic engineering reviews **18**: 149-170.
- VY, B., Z. NI and E. AA (1998). "Tetrapyrroles: diversity, biosynthesis, and biotechnology (review)." Applied Biochemistry and Microbiology **34**(1): 1-18.

Reviewers' Comments:

Reviewer #1:

Remarks to the Author:

The authors have answered the questions properly and revised the manuscript according to my suggestions. It's a nice work. Now, I suggest that this revised manuscript could be accepted.

Reviewer #2:

Remarks to the Author:

I satisfy the answers from the authors. Even for the over requested the new data and discussions on the manuscript or comments by reviewers, the authors are accepted sincerely and answered with respect and made the revised manuscript which includes new data, references and discussions for the comments.